# *daf-16*/FoxO promotes gluconeogenesis and trehalose synthesis during starvation to support survival

Jonathan D Hibshman[1,2], Alexander E Doan[1], Brad T Moore[1], Rebecca EW Kaplan[1,2], Anthony Hung[1], Amy K Webster[1,2], Dhaval P Bhatt[3], Rojin Chitrakar[1], Matthew D Hirschey[3,4,5], L Ryan Baugh[1,2]*

[1]Department of Biology, Duke University, Durham, United States; [2]University Program in Genetics and Genomics, Duke University, Durham, United States; [3]Duke Molecular Physiology Institute, Duke University, Durham, United States; [4]Department of Medicine, Duke University, Durham, United States; [5]Department of Pharmacology & Cancer Biology, Duke University, Durham, United States

**Abstract** *daf-16*/FoxO is required to survive starvation in *Caenorhabditis elegans*, but how *daf-16*/FoxO promotes starvation resistance is unclear. We show that *daf-16*/FoxO restructures carbohydrate metabolism by driving carbon flux through the glyoxylate shunt and gluconeogenesis and into synthesis of trehalose, a disaccharide of glucose. Trehalose is a well-known stress protectant, capable of preserving membrane organization and protein structure during abiotic stress. Metabolomic, genetic, and pharmacological analyses confirm increased trehalose synthesis and further show that trehalose not only supports survival as a stress protectant but also serves as a glycolytic input. Furthermore, we provide evidence that metabolic cycling between trehalose and glucose is necessary for this dual function of trehalose. This work demonstrates that *daf-16*/FoxO promotes starvation resistance by shifting carbon metabolism to drive trehalose synthesis, which in turn supports survival by providing an energy source and acting as a stress protectant.
DOI: https://doi.org/10.7554/eLife.30057.001

*For correspondence:
ryan.baugh@duke.edu

Competing interests: The authors declare that no competing interests exist.

## Introduction

Nutrient availability naturally fluctuates, and animals have a variety of physiological responses that allow them to cope with nutrient stress. Starvation resistance is critical to evolutionary fitness, and fasting is an important clinical intervention with broad-ranging effects on aging and disease (*Longo and Mattson, 2014*). The roundworm *Caenorhabditis elegans* often faces starvation in the wild (*Félix and Braendle, 2010*), and it has developmental adaptations that facilitate starvation survival at multiple points in its lifecycle (*Angelo and Van Gilst, 2009*; *Baugh, 2013*; *Golden and Riddle, 1984*; *Johnson et al., 1984*; *Schindler et al., 2014*). In particular, dauer larvae develop as an alternative to the third larval stage in response to high population density and limited food (*Golden and Riddle, 1984*). Notably, dauer larvae form in anticipation of starvation (food is actually required for dauer development), provisioning fat and developing morphological modifications that support survival. In contrast, worms that hatch in the absence of food (*Escherichia coli* in the laboratory) remain in a state of developmental arrest known as 'L1 arrest' (or 'L1 diapause') until they feed (*Baugh, 2013*). Unlike dauer larvae, arrested L1 larvae have no morphological modification and are provisioned maternally. *C. elegans* L1 arrest provides a powerful organismal model to investigate gene regulatory and metabolic mechanisms that enable animals to cope with acute starvation.

Insulin-like signaling is a critical regulator of starvation survival during L1 arrest (*Baugh and Sternberg, 2006*; *Muñoz and Riddle, 2003*). Feeding promotes insulin-like signaling through the receptor

**eLife digest** Most animals rarely have access to a constant supply of food, and so have evolved ways to cope with times of plenty and times of shortage. Insulin is a hormone that travels throughout the body to signal when an animal is well fed. Insulin signaling inhibits the activity of a protein called FoxO, which otherwise switches on and off hundreds of genes to control the starvation response.

The roundworm, *Caenorhabditis elegans*, has been well studied in the laboratory, and often has to cope with starvation in the wild. These worms can pause their development if no food is available, or divert to a different developmental path if they anticipate that food will be short in future. As with more complex animals, the worm responds to starvation by reducing insulin-like signaling, which in turn activates a FoxO protein called daf-16. When the worms stop feeding, daf-16 is switched on, which is crucial for survival.

It was known how daf-16 stops the roundworm's development, but it was not known how it helps the worms to survive starvation. Now, Hibshman et al. have compared normal roundworm larvae to larvae that are missing the gene for daf-16 to determine how this protein influences the roundworm's ability to survive starvation.

The worms were examined with and without food, to look for which genes were switched on and off by daf-16 during starvation. This revealed that daf-16 controls metabolism, activating a metabolic shortcut that makes the worms produce glucose and begin turning it into another type of sugar, called trehalose. This sugar usually promotes survival in conditions where water is limiting, like dehydration and high salt, but it can also be broken down to release energy. The levels of trehalose in the worms rose within hours of the onset of starvation.

To confirm the importance of trehalose in surviving starvation, roundworms with mutations in genes involved in glucose or trehalose production were examined, as was the effect of giving starving worms glucose or trehalose. Disrupting the production of sugars caused the worms to die sooner of starvation, while supplementing with sugar had the opposite effect meaning the worms survived for longer.

Taken together, these findings reveal that daf-16 protects against starvation by shifting metabolism towards the production of trehalose. This helps worms to survive by both protecting them from stress and providing them with a source of energy. These findings not only extend the current understanding of how animals respond to starvation, but could also lead to improved understanding of diseases where this response goes wrong, including diabetes and obesity.

DOI: https://doi.org/10.7554/eLife.30057.002

daf-2/InsR (**Chen and Baugh, 2014**), which antagonizes the transcription factor daf-16/FoxO (**Lin et al., 1997**; **Ogg et al., 1997**). In the absence of insulin-like signaling, daf-16 is active and promotes developmental arrest and starvation survival (**Baugh and Sternberg, 2006**). daf-16 promotes developmental arrest by inhibiting the dbl-1/TGF-β and daf-12 steroid hormone receptor pathways, but these pathways do not affect starvation survival (**Kaplan et al., 2015**; **Lee and Ashrafi, 2008**). That is, the effects of daf-16 on developmental arrest and starvation resistance are distinct, and how daf-16 promotes starvation resistance is unknown. daf-16 also promotes longevity in fed adults (**Kenyon et al., 1993**), and understanding how it does so has received considerable attention. A variety of studies have identified daf-16 target genes in the context of aging (**Dong et al., 2007**; **Kaletsky et al., 2016**; **Lee et al., 2003**; **Murphy et al., 2003**), resulting in identification of over 3000 genes affected directly or indirectly by daf-16 (**Tepper et al., 2013**). However, these experiments typically examined the effect of constitutive daf-16 activation in fed daf-2/InsR mutant adults rather than conditional effects of daf-16 in response to nutrient availability. We examined the effect of daf-16 on gene expression in L1 larvae starved for ~12 hr (**Kaplan et al., 2015**), but this is well after the starvation response is mounted (**Baugh et al., 2009**), and this experiment did not include nutrient availability as a factor. Consequently, the immediate-early targets of daf-16 involved in the conditional response to starvation have not been identified.

Metabolic adaptations in dauer larvae represent a 'microaerobic' metabolism, with reduced reliance on respiration and oxidative phosphorylation, instead oxidizing fatty acids as fuel for cellular

maintenance (*Braeckman, 2009*; *Burnell et al., 2005*; *O'Riordan and Burnell, 1989*; *1990*; *Wadsworth and Riddle, 1989*). Expression analysis suggests that dauer larvae are metabolically similar to long-lived *daf-2*/InsR mutant adults, with expression of enzymes involved in the glyoxylate shunt (a 'shortcut' through the TCA cycle), gluconeogenesis, and trehalose synthesis upregulated in both (*Depuydt et al., 2014*; *Fuchs et al., 2010*; *McElwee et al., 2004*; *2006*; *Penkov et al., 2015*). However, the acute starvation response that occurs during L1 arrest or other non-dauer stages has not been compared to dauer larvae. Given the unique features of dauer larvae, it is unclear whether their metabolic adaptations represent a universal starvation response.

Trehalose is a disaccharide of glucose formed by an α–α, 1–1 glycosidic bond. Trehalose buffers unicellular and multicellular organisms from osmotic stress, desiccation, heat stress, and freezing, and it can improve proteostasis (*Behm, 1997*; *Elbein et al., 2003*; *Elliott et al., 1996*; *Erkut et al., 2011*; *François et al., 2012*; *Honda et al., 2010*; *Jain and Roy, 2009*; *Lamitina and Strange, 2005*; *Newman et al., 1993*; *Page-Sharp et al., 1999*; *Singer and Lindquist, 1998*; *Tapia and Koshland, 2014*; *Tapia et al., 2015*; *Yoshida et al., 2016*). Trehalose is not synthesized in vertebrates, but it confers desiccation tolerance on human cells (*Guo et al., 2000*). In *C. elegans*, simultaneous disruption of both trehalose 6-phosphate synthase genes (*tps-1* and *tps-2*) reduces desiccation tolerance in dauer larvae (*Erkut et al., 2011*). The glyoxylate shunt also supports desiccation tolerance in dauer larvae (*Erkut et al., 2016*). However, regulation of these metabolic adaptations is not understood. Furthermore, it remains unclear what role trehalose or trehalose synthesis plays in mediating starvation resistance.

The protective role of trehalose in conditions where water is limiting is attributed to its function as a compatible solute and its ability to replace water molecules at the surface of proteins and lipid bilayers, stabilizing polar interactions and preserving membrane organization and protein structure (*Erkut et al., 2011*, *2012*; *Leekumjorn and Sum, 2008*). We refer to this biochemical mechanism of trehalose function as that of a 'stress protectant'. However, trehalose can also be used as an energy source to fuel glycolysis. For example, trehalose is the primary circulating sugar in insects, satisfying the high-energy demands of flight (*Clegg and Evans, 1961*; *Wyatt and Kale, 1957*). Given multiple reported physiological roles of trehalose and variation across taxa, there is debate over the relative importance of the different roles of trehalose in different contexts (*Crowe, 2007*; *Hohmann et al., 1996*).

We used a combination of genome-wide expression analysis and metabolomics to determine the nutrient-dependent effects of *daf-16*/FoxO in recently hatched L1-stage larvae of *C. elegans*. We report that during acute starvation, *daf-16* promotes carbon flux through the glyoxylate shunt and gluconeogenesis towards trehalose synthesis, similar to the metabolic adaptations that occur in dauer larvae. Furthermore, we demonstrate that this metabolic shift is physiologically significant and supports starvation resistance. Multiple lines of evidence show that trehalose promotes starvation survival through two distinct mechanisms: as a stress protectant without being catabolized and as an energy source for glycolysis. We also show that trehalose and glucose interconvert and that interconversion is necessary for the dual function of trehalose. This work elucidates how *daf-16*/FoxO promotes starvation resistance, and it reveals a central role of trehalose metabolism in maintaining organismal energy homeostasis and stress resistance during acute starvation.

## Results

### *daf-16*/FoxO promotes the glyoxylate shunt, gluconeogenesis, and trehalose synthesis

We performed genome-wide expression analysis to determine the immediate effects of *daf-16*/FoxO activity in L1-stage larvae upon starvation. We used a 'double-bleach' procedure to ensure synchronized hatching (*Baugh, 2013*; *Baugh et al., 2009*). We measured expression in wild-type (WT) and *daf-16* mutant larvae shortly after hatching with and without food (*E. coli* HB101, *Figure 1—figure supplement 1*). These data compare favorably with a published time series of WT gene expression using the same staging procedure (*Baugh et al., 2009*), confirming a robust effect of nutrient availability and that the samples represent recently hatched larvae (*Figure 1—figure supplement 2A*). Nutrient availability had a substantially larger effect on mRNA expression than genotype, with ~4000 genes displaying differential expression between fed and starved conditions in

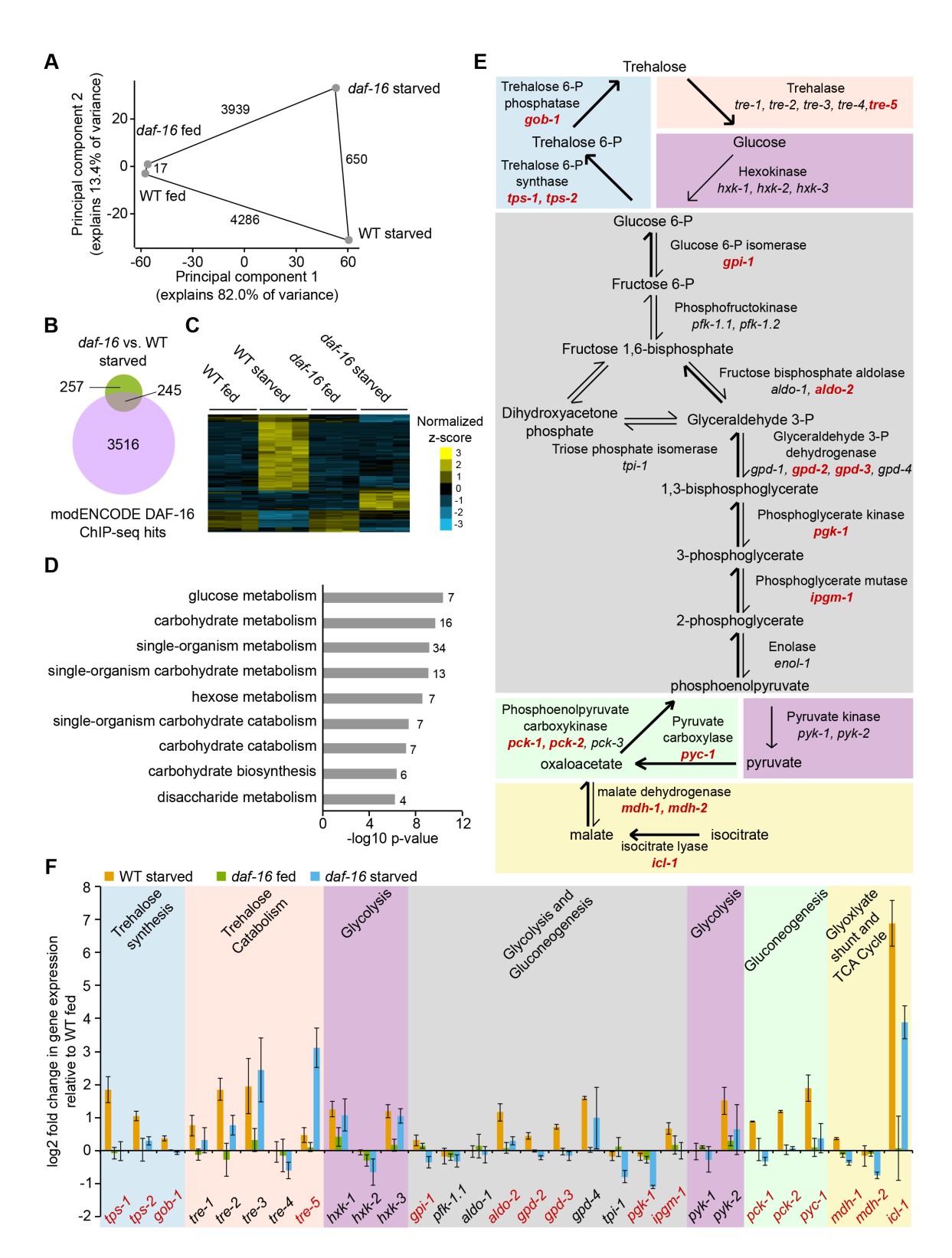

**Figure 1.** *daf-16*/FoxO regulates expression of carbohydrate metabolism genes in response to starvation. (**A**) Principal component analysis (PCA) of 5177 genes differentially expressed in any of four pairwise comparisons or with a significant interaction between genotype and condition (FDR < 0.05). *Figure 1 continued on next page*

*Figure 1 continued*

The number of differentially expressed genes in each pairwise comparison is shown along the line connecting conditions. Mean expression values from three independent biological replicates were used. (B) Overlap of genes differentially expressed during starvation in *daf-16* mutants compared to wild-type (WT) and targets of DAF-16 binding from modEncode ChIP-seq data (annotated in *Tepper et al., 2013*). (C) Genes with a significant interaction between genotype and condition (FDR<0.05, n=103) were hierarchically clustered. Genes along the y-axis are colored in each condition based on their z-score and normalized to the average across all conditions. (D) Gene Ontology (GO) terms enriched among 103 genes that display a significant interaction between genotype and condition (FDR<0.05) are plotted by −log10 p-value. The number of genes for each GO term is listed to the right of each bar. (E) A schematic representation of carbohydrate metabolism based on *McElwee et al. (2006)*. Differentially expressed genes in *daf-16* compared to WT during starvation are in red (adjusted FDR < 0.05). (F) Log2 gene expression relative to fed WT is plotted for genes depicted in D, with the exception of *pfk-1.2*, *gpd-1*, *enol-1*, and *pck-3*, each of which did not have consistently detectable expression levels. Means and SDs of three biological replicates are shown.

DOI: https://doi.org/10.7554/eLife.30057.003

The following source data, source code and figure supplements are available for figure 1:

**Source code 1.** Code used for analysis of microarray experiment.
DOI: https://doi.org/10.7554/eLife.30057.006
**Source data 1.** Data compiled from microarray analysis of fed and starved WT and *daf-16* mutant worms.
DOI: https://doi.org/10.7554/eLife.30057.007
**Figure supplement 1.** Gene expression of fed and starved WT and *daf-16* larvae.
DOI: https://doi.org/10.7554/eLife.30057.004
**Figure supplement 2.** Gene expression analysis of fed and starved larvae.
DOI: https://doi.org/10.7554/eLife.30057.005

both WT and *daf-16* mutants (*Figure 1A*; false-discovery rate [FDR]<0.05). Given the *daf-16* mutant phenotypes during L1 starvation (starvation-sensitive and arrest-defective), it was surprising that *daf-16* mutants did not have a larger effect on the starvation response (*Figure 1—figure supplement 1*), with only 650 genes affected in starved larvae (*Figure 1A*). That is, the starvation response was largely intact in *daf-16* mutants (*Figure 1—figure supplement 1*, *Figure 1—figure supplement 2A, B*). This result demonstrates that other pathways are critical for the starvation response, and it is consistent with capturing relatively early, direct effects of *daf-16*. Indeed, DAF-16 binds approximately half of the genes affected by it during starvation based on modENCODE ChIP-seq results (hypergeometric enrichment p=3.3E-8) (*Niu et al., 2011*) (*Figure 1B*), suggesting direct regulation of these genes. Notably, *daf-16* had much less of an effect on fed larvae, affecting only 17 genes (*Figure 1A*). This is consistent with *daf-16* functioning in starved but not fed larvae, where it is localized to the cytoplasm due to antagonism from insulin-like signaling (*Weinkove et al., 2006*). Moreover, the differential effect of *daf-16* in fed and starved larvae confirms that we captured the effects of conditional regulation by *daf-16*.

Expression analysis suggests that *daf-16*/FoxO has a pervasive effect on central carbon metabolism in response to starvation. We used a two-factor analysis to formally identify genes with a significant interaction between condition (fed vs. starved) and genotype (WT vs. *daf-16*). This analysis addresses our experimental design explicitly, but it has less statistical power than a pairwise test, and it identified only 103 genes that are regulated by nutrient availability in *daf-16*-dependent fashion (FDR < 0.05). As a positive control, this stringent statistical analysis identified the superoxide dismutase gene *sod-3*, a known direct target of DAF-16 (*Henderson et al., 2006*; *Zhang et al., 2013*) (*Figure 1—figure supplement 2C*, interaction FDR = 0.002). Like *sod-3*, the majority of these 103 genes were upregulated by starvation in WT, but not in *daf-16* mutants (*Figure 1C*). Gene ontology (GO) term enrichment analysis for these 103 genes revealed substantial bias towards metabolism terms, particularly carbohydrate metabolism (*Figure 1D*). A schematic representation of carbohydrate metabolism shows 16 enzymes that are differentially expressed between WT and *daf-16* during starvation (*Figure 1E* in red). Notably, 15 of these differentially expressed genes were downregulated in *daf-16* mutants (all but *tre-5*), and ChIP-seq suggests that DAF-16 binds each of them (*Niu et al., 2011*), suggesting DAF-16 directly activates transcription of these metabolic genes during starvation. Time-series analysis revealed variation in dynamics of these *daf-16*-regulated metabolic enzymes but broadly confirmed sustained upregulation during L1 starvation (*Figure 1—figure supplement 2D*) (*Baugh et al., 2009*).

*daf-16*/FoxO appears to promote metabolic flux through the glyoxylate shunt, gluconeogenesis, and trehalose synthesis in response to starvation. *icl-1*, the gene encoding the isocitrate lyase/malate synthase enzyme essential for the glyoxylate shunt, was upregulated during starvation in *daf-16*-dependent fashion (*Figure 1F*). Many glycolysis enzymes are bidirectional and also catalyze the reverse gluconeogenic reactions, but several genes are exclusive to glycolysis or gluconeogenesis, allowing us to infer how gene expression changes affect carbon flux. Hexokinase and pyruvate kinase catalyze unidirectional glycolytic reactions, and none of the five genes encoding these two enzymes were affected by *daf-16*. In contrast, the gluconeogenic enzymes pyruvate carboxylase (*pyc-1*) and phosphoenolpyruvate carboxykinase (PEPCK, *pck-1* and *pck-2*) were upregulated during starvation in WT, but not in *daf-16* mutants. Trehalose synthesis genes *tps-1*, *tps-2*, and *gob-1* were also upregulated during starvation in *daf-16*-dependent fashion (*Figure 1F*). Collectively, these results suggest that increased flux through the glyoxylate shunt and gluconeogenesis provides glucose for trehalose synthesis.

*daf-16*/FoxO-dependent changes in metabolic gene expression during L1 starvation translate into changes in carbon metabolism. We conducted targeted metabolomics for panels of amino acids, organic acids, and acyl carnitines using the same experimental design as for expression analysis. Principal component analysis of these data showed that metabolic profiles are significantly affected by nutrient availability in WT, but the difference between conditions is reduced in *daf-16* mutants (*Figure 2A*), consistent with *daf-16* activity contributing to the difference between conditions in WT. However, none of the individual metabolites targeted for analysis was significantly affected by genotype during starvation after correction for multiple testing (*Figure 2—figure supplement 1*). We suspect our inability to detect significant individual differences is due to lack of statistical power since an apparent effect on the overall metabolic profile was observed (*Figure 2A*). In contrast, non-targeted metabolomic analysis revealed a 5.7-fold increase in disaccharide levels during starvation in WT but a mere 1.8-fold increase in *daf-16* mutants (*Figure 2B*). Because expression of trehalose synthesis genes was increased during starvation in *daf-16*-dependent fashion (*Figure 1*), we suspected this peak was due to increased trehalose levels. Indeed, specifically measuring trehalose in a biochemical assay revealed a 3.9-fold increase in WT during starvation and a mere 1.5-fold increase in *daf-16* mutants (*Figure 2C*). Time-series analysis revealed a difference in trehalose levels between fed and starved larvae within 4 hr of hatching (*Figure 2D*). These results demonstrate that the metabolic gene expression changes caused by *daf-16* during L1 starvation are physiologically significant, with the net effect of increasing steady-state levels of trehalose.

## The glyoxylate shunt and gluconeogenesis support starvation survival

The metabolic shift mediated by *daf-16*/FoxO promotes starvation resistance. We measured L1 starvation survival of mutants that specifically affect glycolysis, gluconeogenesis, and the glyoxylate shunt (*Figure 3A*, *Supplementary file 1*). *daf-16* mutants were extremely sensitive to starvation, as expected (*Baugh and Sternberg, 2006*). Consistent with our hypothesis that gluconeogenesis contributes to starvation survival, the phosphoenolpyruvate carboxykinase mutant *pck-1* was sensitive to starvation, though not to the same extent as *daf-16*. The glyoxylate shunt is disrupted in *icl-1*/isocitrate lyase/malate synthase mutants, and *icl-1* mutants were also starvation-sensitive, consistent with the glyoxylate shunt feeding into gluconeogenesis during starvation. In contrast, we hypothesized that glycolysis is less important to starvation survival than gluconeogenesis based on *daf-16*-dependent changes in gene expression (*Figure 1*). Indeed, survival of pyruvate kinase (*pyk-1*) mutants, which specifically affect glycolysis, was indistinguishable from WT (*Figure 3A*, *Supplementary file 1*). These results are consistent with the glyoxylate shunt and gluconeogenesis being particularly important to starvation survival.

Glycolysis is relatively more important to fed larvae than starved larvae. We measured size (length) after 48 hr of larval growth in well-fed metabolic mutants (*Figure 3B*). In contrast to their effect on starvation survival, *daf-16* and *icl-1* mutants did not have a significant effect on growth rate. These results are consistent with *daf-16* and the glyoxylate shunt being relatively starvation-specific. However, *pck-1* and *pyk-1* mutants grew relatively slow. These results suggest that fed larvae rely more on glycolysis than starved larvae, and that gluconeogenesis is important to fed and starved larvae. Furthermore, like *daf-16,* the differential effects of *pyk-1* and *icl-1* on starvation survival and larval growth suggest their phenotypes are not simply due to general sickness.

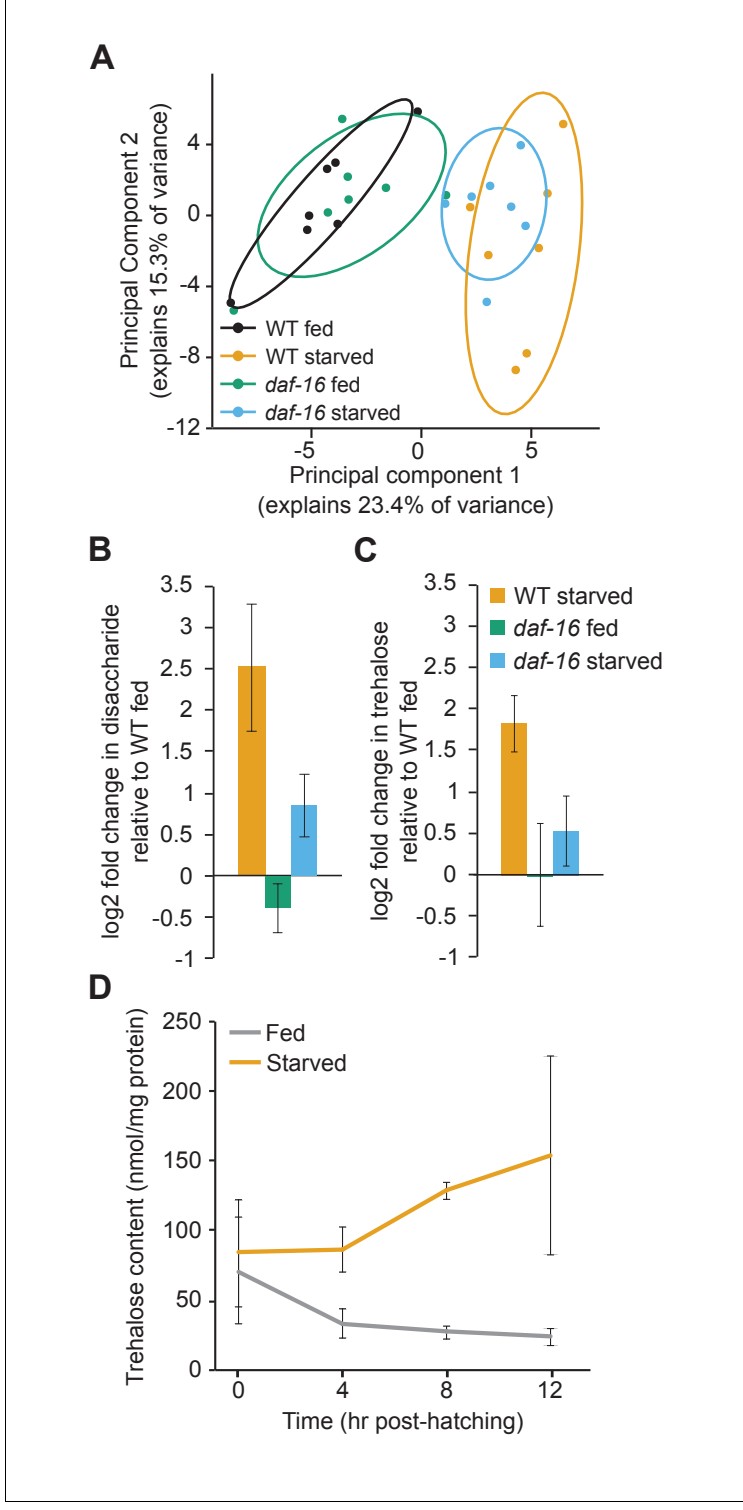

**Figure 2.** *daf-16*/FoxO shifts metabolic flux and increases trehalose synthesis during starvation. (**A**) Principal component analysis (PCA) of targeted metabolomic analysis fed and starved wild-type (WT) and *daf-16* L1 larvae shows separation by condition, but not by genotype. 80% confidence ellipses are included with seven biological replicates. (**B**) Disaccharide levels were measured in non-targeted metabolomic analysis. Mean and SD of log2 fold-change relative to fed WT is plotted for four biological replicates ($p_{int}$=0.05, n = 4, two-way ANOVA). (**C**) Trehalose levels were measured in a biochemical assay, and log2 fold-change relative to fed WT is plotted for four biological replicates ($p_{int}$=0.003, n = 4, two-way ANOVA). (**D**) Trehalose content in fed (gray) and starved (orange)
*Figure 2 continued on next page*

*Figure 2 continued*

worms is plotted over time. Fed and starved conditions are significantly different (p=0.01, n = 2, two-way ANOVA). Mean and standard error of the mean of two biological replicates are shown.

DOI: https://doi.org/10.7554/eLife.30057.008

The following source data and figure supplement are available for figure 2:

**Source data 1.** Raw data for display items in *Figure 2*.

DOI: https://doi.org/10.7554/eLife.30057.010

**Figure supplement 1.** Metabolomic analysis of WT and *daf-16*/FoxO worms during feeding and starvation.

DOI: https://doi.org/10.7554/eLife.30057.009

---

Pharmacological analysis corroborates the results of genetic analysis of metabolic mutants. We assayed the effect of the glycolytic inhibitor 2-deoxy-D-glucose (2-DG) in starved and fed larvae, measuring survival and growth rate, respectively. Dose–response curves for starvation survival and growth rate are clearly distinct (*Figure 3C*, *Supplementary file 1*). Doses of 2-DG up to 20 mM had no effect on starvation survival (p=0.84, unpaired t-test, n = 4) but inhibited growth (p=0.0002, unpaired t-test, n = 3). These results are consistent with larvae relying more on glycolysis for growth when fed than survival when starved. 3-mercaptopicolinic acid (3 MPA) is an inhibitor of PEPCK and has been shown to disrupt gluconeogenesis in plants and mammals (*DiTullio et al., 1974*; *Ray and Black, 1976*), and we expect it to do the same in *C. elegans*. 3 MPA reduced starvation survival and growth comparably (*Figure 3D*). Similarity in 3-MPA dose–response curves corroborates results with *pck-1* mutants (*Figure 3A,B*), suggesting that gluconeogenesis is relatively important in fed and starved larvae. In summary, genetic and pharmacological analyses suggest that a shift in metabolism away from glycolysis toward the glyoxylate shunt and gluconeogenesis is a physiologically significant aspect of adaptation to starvation.

## Trehalose supports starvation survival

Trehalose synthesis promotes starvation survival. Trehalose-6-phosphate synthase catalyzes the first step of trehalose synthesis by converting glucose-6-phosphate and UDP-glucose into trehalose 6-phosphate, and two genes (*tps-1* and *tps-2*) encode this enzyme in *C. elegans* (*Pellerone et al., 2003*). *tps-1* mutants were starvation-sensitive, and *tps-2* mutants were marginally sensitive (p=0.07, *Figure 4A*). A *tps-1; tps-2* double mutant was also starvation-sensitive, but no more sensitive than the *tps-1* single mutant. These results suggest that elevated levels of trehalose, or trehalose synthesis itself, supports starvation survival.

Reporter gene analysis confirmed transcriptional upregulation of *tps-1* during L1 starvation and suggested that induction occurs in the intestine. Using promoter–GFP fusions (*McKay et al., 2003*), we observed significant upregulation of P*tps-1*::GFP in whole L1 larvae when starved compared to fed (*Figure 4B–D*). In contrast, upregulation of P*tps-2*::GFP during starvation was not significant in whole larvae (*Figure 4E–G*). Lack of significance may be due to absence of regulatory elements in the reporter gene or whole-worm analysis, with tissue-specific differences being obscured. Indeed, each reporter was visible primarily in hypodermis and neurons in fed L1 larvae, and expression appeared to be induced specifically in the intestine in response to starvation (*Figure 4B–F*; *Figure 4—figure supplement 1*). P*tps-1*::GFP had at least some visible intestinal expression in 75% of fed larvae but clear intestinal expression in 100% of starved larvae, with apparently increased intensity (n = 44 and 31 worms, respectively). Likewise, P*tps-2*::GFP was visible in the intestine of 19% of fed larvae and 90% of starved larvae (n = 32 and 30 worms, respectively). These results suggest trehalose synthesis is upregulated in the intestine of starved L1 larvae.

Both *tps-1* and *tps-2* contribute to trehalose synthesis in starved L1 larvae. Each single mutant had significantly reduced trehalose content with some residual trehalose (*Figure 4H*). In contrast, trehalose levels were at background in the *tps-1; tps-2* double mutant, which was significantly different from each single mutant. These results show that *tps-1* and *tps-2* both contribute to trehalose synthesis during L1 arrest (*Figure 4A*).

Supplementation of otherwise starved L1 larvae with trehalose in the buffer increases survival substantially. 1 mM (not shown) and 10 mM supplementation did not affect survival, but 20 mM significantly increased survival with a clear dose response that reaches a maximum around 50 mM

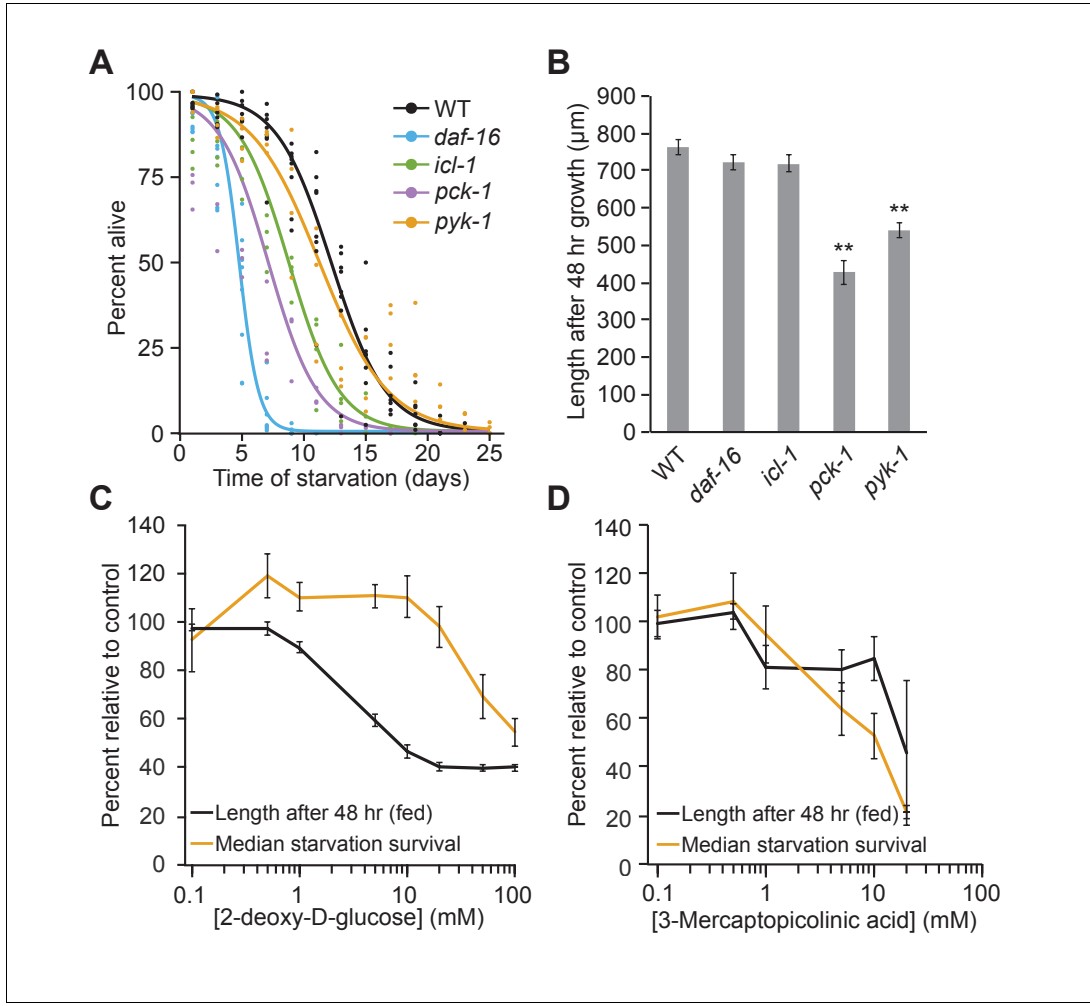

**Figure 3.** Fed and starved larvae differ in their reliance on the glyoxylate shunt and glycolysis. (**A**) Starvation survival curves are plotted for wild-type (WT), *daf-16, icl-1*, and *pck-1* mutants. Here and throughout, logistic regression is used to fit survival curves. See **Supplementary file 1** for median survival and statistics. (**B**) Worm length after 48 hr of feeding on *Escherichia coli* OP50 on plates is plotted for WT, *daf-16, icl-1*, and *pck-1* mutants. Mean and standard error of the mean of three biological replicates is shown. *pck-1* and *pyk-1* were significantly different than WT (p=0.001, p=0.002, unpaired t-test, n = 3). *daf-16* or *icl-1* were not different than WT (p=0.23, p=0.24, unpaired t-test, n = 3). **p<0.01 compared with WT. (**C**) Normalized dose–response curves for 2-deoxy-d-glucose in fed and starved larvae are plotted. (**D**) Normalized dose –response curves for 3-mercaptopicolinic acid (3 MPA) in fed and starved larvae are plotted. (**C and D**) Larvae were fed HB101 *E. coli* lysate in liquid culture and their length was measured after 48 hr (black) or they were starved and survival medians were determined from logistic regressions of survival curves (orange). These data are normalized within each replicate as a percentage of control (no drug) animals. The x-axis is log transformed.
DOI: https://doi.org/10.7554/eLife.30057.011

The following source data is available for figure 3:

**Source data 1.** Raw data for display items in **Figure 3**.
DOI: https://doi.org/10.7554/eLife.30057.012

---

(**Figure 4I**, **Supplementary file 1**). 50 mM and higher concentrations of trehalose doubled survival during L1 arrest. The dose response for survival was mirrored by measurement of internal levels of trehalose after supplementation, also reaching a maximum around 50 mM (**Figure 4J**). This result confirms that larvae can internalize trehalose from the buffer, and it suggests that saturation of internalization limits the effect of supplementation on survival. L1 larvae decrease in length during starvation, but worms supplemented with 50 mM trehalose are longer than control worms after 8 d of

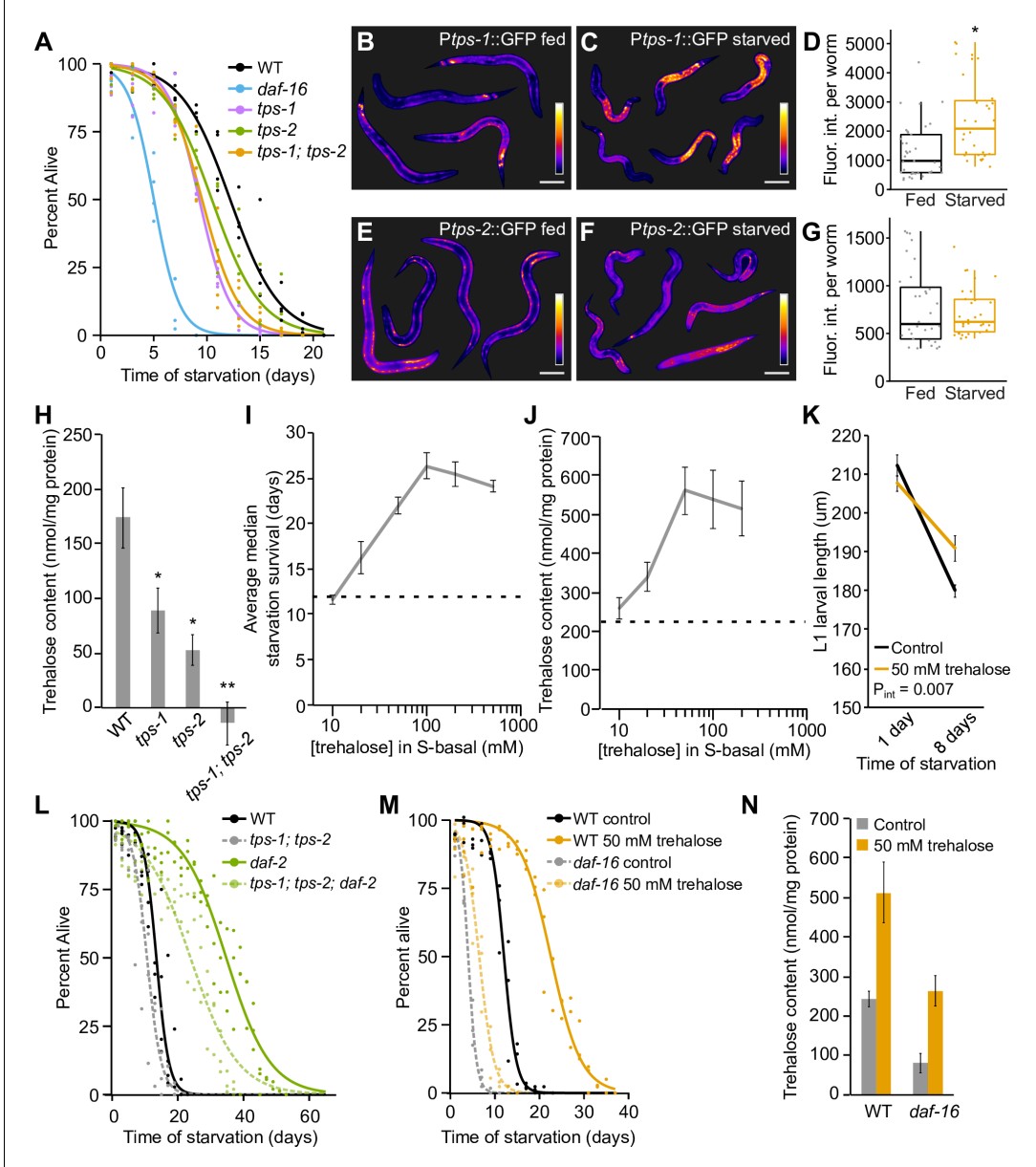

**Figure 4.** Trehalose supports starvation survival and regulation of its synthesis partially explains the effects of insulin-like signaling on starvation resistance. (A) Starvation survival curves are plotted for wild-type (WT), *tps-1*, *tps-2*, *tps-1; tps-2*, and *daf-16*. (B and C,E and F) Representative P*tps-1*::GFP (BC14885) and P*tps-2*::GFP (BC14876) worms either fed or starved for ~12 hr are shown with false coloring to depict relative fluorescent intensity according to the calibration bar. Scale bars are 50 μm. (D and G) Boxplots indicate the distribution of average pixel intensity per worm in each reporter background in fed and starved worms. Each point represents an individual worm sampled from one of three biological replicates. P*tps-1*::GFP was significantly brighter during starvation (p=0.04, unpaired t-test, n = 3). P*tps-2*::GFP intensity was not different between fed and starved worms (p=0.50, unpaired t-test, n = 3). (H) Trehalose levels of WT, *tps-1*, *tps-2*, and *tps-1; tps-2* are plotted. *tps-1* and *tps-2* each had reduced levels of trehalose compared to WT (p=0.05, p=0.01, unpaired t-test, n = 4). The double mutant was significantly different from WT and each single mutant (p=0.001, p=0.01, p=0.03, unpaired t-test, n = 4). (I) Median survival time is plotted for a worms exposed to a range of concentrations of trehalose in the buffer in which worms are starved. The dashed line indicates median survival of control animals. (J) Trehalose content 1 d after hatching is plotted for worms exposed to a range of concentrations of trehalose in S-basal (n = 12). The dashed line indicates trehalose content of control animals. (K) L1 larval size is plotted for WT worms with and without trehalose supplementation. Worms starved for 8 d are shorter than worms starved for 1 d (p=1.3E-05, unpaired t-test, n = 5). Worms supplemented with trehalose remain larger after 8 d of starvation (p_int=0.007, two-way ANOVA, n = 5). (L) Starvation survival curves are plotted for WT, *tps-1; tps-2* (four biological replicates), *daf-2*, and *tps-1; tps-2; daf-2* (three biological replicates). There is a significant interaction by two-way ANOVA between median survivals (p_int = 0.02). (M) Starvation survival curves for WT and *daf-16* with and without 50 mM trehalose supplementation. (N) Trehalose content is plotted for N2 and *daf-16* with and without 50 mM trehalose supplementation (n = 7). Mean and standard error of the mean of biological replicates are plotted in H–K and N.

*Figure 4 continued on next page*

*Figure 4 continued*

DOI: https://doi.org/10.7554/eLife.30057.013

The following source data and figure supplement are available for figure 4:

**Source data 1.** Raw data for display items in *Figure 4*.

DOI: https://doi.org/10.7554/eLife.30057.015

**Figure supplement 1.** *tps-1* and *tps-2* are primarily expressed in hypodermis and intestine during starvation.

DOI: https://doi.org/10.7554/eLife.30057.014

starvation (*Figure 4K*). Reduction in the rate of shrinkage during starvation further supports the conclusion that trehalose supports starvation resistance.

Changes in insulin-like signaling act through trehalose synthesis to affect starvation survival. L1 starvation survival analysis of *tps-1; tps-2; daf-2* triple mutants revealed quantitative epistasis between *tps-1; tps-2* and *daf-2* (*Figure 4L*, *Supplementary file 1*, two-way ANOVA $p_{int}$ = 0.02). This genetic interaction is consistent with *daf-2*/InsR mutants increasing starvation resistance by upregulating trehalose synthesis via DAF-16. Likewise, supplementing starvation-sensitive *daf-16*/FoxO mutants with 50 mM trehalose significantly increased survival, but not to the extent of WT (*Figure 4M*, *Supplementary file 1*, two-way ANOVA $p_{int}$ < 0.0001). Supplementation of *daf-16* mutants with 50 mM trehalose increased internal trehalose levels to WT levels without supplementation (*Figure 4N*), despite supplementation incompletely restoring survival of *daf-16* (*Figure 4M*). This result together with incomplete suppression of *daf-2* starvation resistance by *tps-1; tps-2* (*Figure 4L*) indicates that insulin-like signaling affects more than just trehalose synthesis to regulate starvation resistance.

## Trehalose functions as a stress protectant and an energy source to support survival

We used a variety of approaches to distinguish between possible physiological roles of trehalose in supporting starvation survival. Trehalose is known to preserve membrane organization and protein structure during various forms of abiotic stress (*Erkut et al., 2011*; *Guo et al., 2000*; *Jain and Roy, 2009*; *Matsuda et al., 2015*; *Singer and Lindquist, 1998*; *Tapia et al., 2015*). It is possible that trehalose functions similarly as a stress protectant during starvation, preserving integrity of proteins, membranes or other cellular components to support survival. It is also possible that trehalose serves as a carbon source during starvation to provide energy via glycolysis in support of survival. Notably, these two hypothetical roles of trehalose are not mutually exclusive.

Catabolism of trehalose and use as an energy source requires trehalase activity. We disrupted trehalase activity to test the relative contributions of trehalose to starvation survival as an energy source and a stress protectant. Reporters for *tre-2*, *tre-3*, *tre-4* and *tre-5* were expressed in relatively distinct patterns (*Figure 5—figure supplement 1A–D*). No individual trehalase mutant significantly altered starvation survival (*Figure 5—figure supplement 1E*, *Supplementary file 1*). We generated a *tre-1; tre-5; tre-2; tre-3;* and *tre-4* quintuple mutant to eliminate all trehalase activity. We confirmed that the quintuple mutant has elevated trehalose levels (*Figure 5A*), consistent with a failure to catabolize trehalose. Surprisingly, starvation survival of the trehalase quintuple mutant was not significantly different than WT (*Figure 5B*, *Supplementary file 1*). However, starvation survival of the quintuple mutant was only partially extended by supplementation with 50 mM trehalose compared to WT (*Figure 5B*, *Supplementary file 1*). This intermediate effect is consistent with a role as stress protectant, but it also suggests that trehalose, at least when supplemented exogenously, must be metabolized in order to maximally support survival.

Supplementation with a non-hydrolyzable form of trehalose corroborated genetic ablation of trehalase activity. In contrast to naturally occurring α–α trehalose, α–β trehalose cannot be hydrolyzed by trehalases but should retain function as a stress protectant. We confirmed that the biochemical assay we use to measure trehalose levels does not detect α–β trehalose (*Figure 5—figure supplement 1F*). In contrast to α–α trehalose, supplementation with 50 mM α–β trehalose did not significantly increase internal α–α trehalose levels (*Figure 5C*), indicating that α–β trehalose is not substantially converted to α–α trehalose in vivo. Nonetheless, supplementation with 50 mM α–β trehalose increased starvation survival (*Figure 5D*). This clearly suggests a role for trehalose as a stress

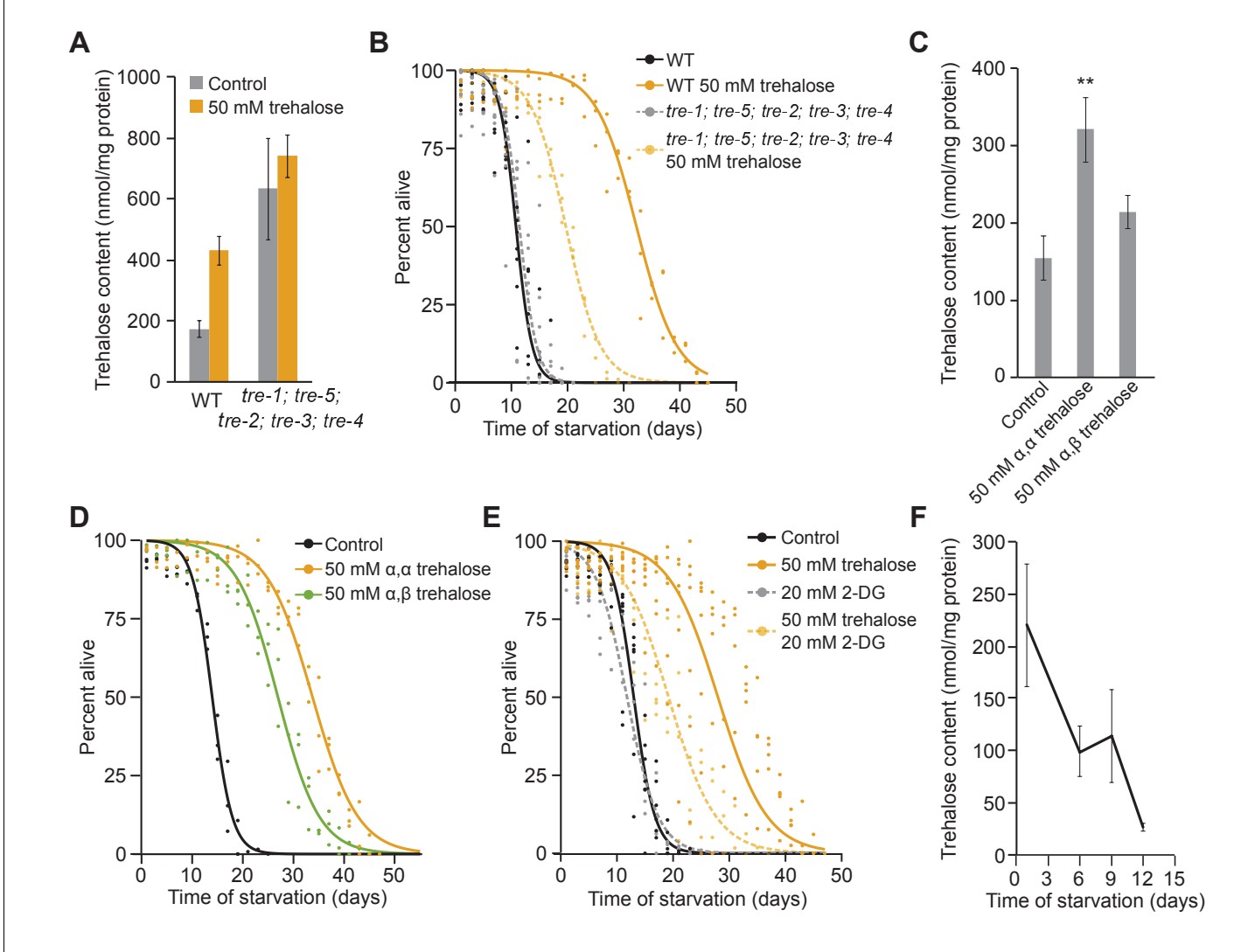

**Figure 5.** Trehalose has dual function as energy source and stress protectant during starvation. (**A**) Trehalose content is plotted for wild-type (WT) and trehalase quintuple mutant with and without 50 mM trehalose supplementation. *tre-1; tre-5; tre-2; tre-3; tre-4* mutants have significantly higher levels of trehalose than WT (p=0.04, unpaired t-test, n = 4). While WT worms have higher corporeal trehalose upon supplementation (p=0.004, unpaired t-test, n = 4), trehalose content does not significantly change in *tre-1; tre-5; tre-2; tre-3; tre-4* mutants (p=0.57, unpaired t-test, n = 4). (**B**) Starvation survival curves are plotted for WT and the trehalase quintuple mutant with and without 50 mM trehalose supplementation. (**C**) Trehalose content is plotted for WT worms with and without supplementation of 50 mM α–α trehalose or 50 mM α-β trehalose (n = 7). Supplementation with 50 mM α–α trehalose increased corporeal trehalose (p=0.007, unpaired t-test), but supplementation with 50 mM α–β trehalose did not significantly affect trehalose levels compared to control worms in S-basal (p=0.13, unpaired t-test). (**D**) Starvation survival curves are plotted for WT with and without supplementation with 50 mM α–α trehalose or 50 mM α–β trehalose. (**E**) Starvation survival curves are plotted for WT with or without 20 mM 2-DG and with or without 50 mM trehalose supplementation. (**F**) Trehalose content in WT worms during starvation is plotted. Endogenous trehalose content decreases over time (p=0.03, ANOVA, n = 4). Mean and standard error of the mean are plotted.

DOI: https://doi.org/10.7554/eLife.30057.016

The following source data and figure supplements are available for figure 5:

**Source data 1.** Raw data for display items in *Figure 5*.
DOI: https://doi.org/10.7554/eLife.30057.018

**Figure supplement 1.** Controls for experiments where trehalose catabolism is prevented.
DOI: https://doi.org/10.7554/eLife.30057.017

**Figure supplement—source data 1.** Raw data for display items in Figure 5-figure supplement 1
DOI: https://doi.org/10.7554/eLife.30057.019

protectant. However, 50 mM α–β trehalose did not increase survival to the same extent as α–α trehalose (*Figure 5D*). Consistent with the results of trehalose supplementation in the trehalase quintuple mutant, this intermediate effect suggests that trehalose functions as a stress protectant but also must be metabolized to maximally support survival.

Trehalose is used for glycolysis to support starvation survival. Inhibiting glycolysis with 20 mM 2-DG did not affect starvation survival of WT (*Figures 3C* and *5E*). However, 20 mM 2-DG reduced survival in worms supplemented with 50 mM trehalose (*Figure 5E*). This intermediate effect of trehalose supplementation in worms with inhibited glycolysis further supports the conclusion that trehalose is used as an energy source during starvation. Metabolism of trehalose would presumably cause a decrease in levels as it is catabolized and resources to synthesize more of it are depleted. Indeed, trehalose levels decreased over time during extended L1 starvation (*Figure 5F*), consistent with it being used as an energy source. In summary, multiple lines of evidence suggest trehalose functions as both a stress protectant and an energy source during L1 starvation.

## Dual function of trehalose requires interconversion of trehalose and glucose

Trehalose supplementation does not fully complement the *tps-1; tps-2* mutant. *tps-1; tps-2* had reduced starvation survival, as expected (*Figure 4A,K*), and trehalose supplementation increased survival (*Figure 6A*). However, survival of *tps-1; tps-2* with supplementation was less than WT with supplementation. Trehalose supplementation also increased heat shock survival on the first day of L1 arrest, and again survival of *tps-1; tps-2* with supplementation was less than WT with supplementation (*Figure 6B*). This discrepancy in survival between genotypes with supplementation was reconciled by measuring corporeal trehalose levels with and without supplementation. The *tps-1; tps-2* mutant had no detectable trehalose (*Figure 6C*), as expected (*Figure 4H*), consistent with it being null for trehalose synthesis activity. 50 mM trehalose supplementation increased WT levels, as expected (*Figure 4N*), but had no effect on *tps-1; tps-2* (*Figure 6C*). This result reveals that trehalose synthesis is required to maintain an internal pool of trehalose even with supplementation, as if ingested and possibly endogenous trehalose is rapidly catabolized during starvation. Given the inability of the *tps-1; tps-2* mutant to maintain significant levels of internal trehalose, we believe that the trehalose supplemented in the medium is used primarily as an energy source but does not itself substantially contribute to survival as a stress protectant. Consequently, supplementation of *tps-1; tps-2* with trehalose produces an intermediate survival effect, analogous to other conditions where its function as a stress protectant and an energy source are uncoupled (*Figure 5*).

Synthesis of trehalose from other sugars supports starvation survival and heat resistance. Supplementation with 50 mM maltose or 100 mM glucose extends starvation survival to the same extent as 50 mM trehalose (*Figure 6A*). 100 mM glucose was used because trehalose and maltose are disaccharides of glucose. Supplementing *tps-1; tps-2* mutants with maltose or glucose also increased starvation survival but not to the same extent as supplementation in WT. Furthermore, maltose and glucose supported heat shock survival, but not to the same extent in *tps-1; tps-2* as WT (*Figure 6B*). We hypothesized these sugars increase survival because they are used to fuel glycolysis and synthesize trehalose, both of which contribute to survival. Indeed, supplementation with glucose increased endogenous trehalose levels in WT, comparable to trehalose supplementation, but not in *tps-1; tps-2* (*Figure 6C*). Maltose supplementation also increased trehalose levels in WT, but to a lesser extent. Thus, we conclude that sugars supplemented in the medium of otherwise starved larvae interconvert between glucose and trehalose, that trehalose synthesis is necessary to maintain high steady-state levels of trehalose even with supplementation, and that in the absence of trehalose (re-) synthesis sugar supplementation contributes to survival primarily as an energy source.

Gene expression analysis corroborates the conclusion that trehalose synthesis is necessary for the full physiological effect of trehalose supplementation. We sequenced mRNA from WT and *tps-1; tps-2* mutants on the first day of L1 starvation with and without 50 mM trehalose supplementation. Notably, the gene expression profile associated with the trehalose synthesis defect of the *tps-1; tps-2* mutant is not fully complemented by trehalose supplementation: 116 genes were differentially expressed in *tps-1; tps-2* compared to WT when both were supplemented with trehalose (*Figure 7A*). 285 genes were differentially expressed in response to trehalose supplementation in WT, but only 99 genes were affected by supplementation in *tps-1; tps-2*. That is, there was a common response to supplementation in the two genotypes, but it was dampened in the mutant so the

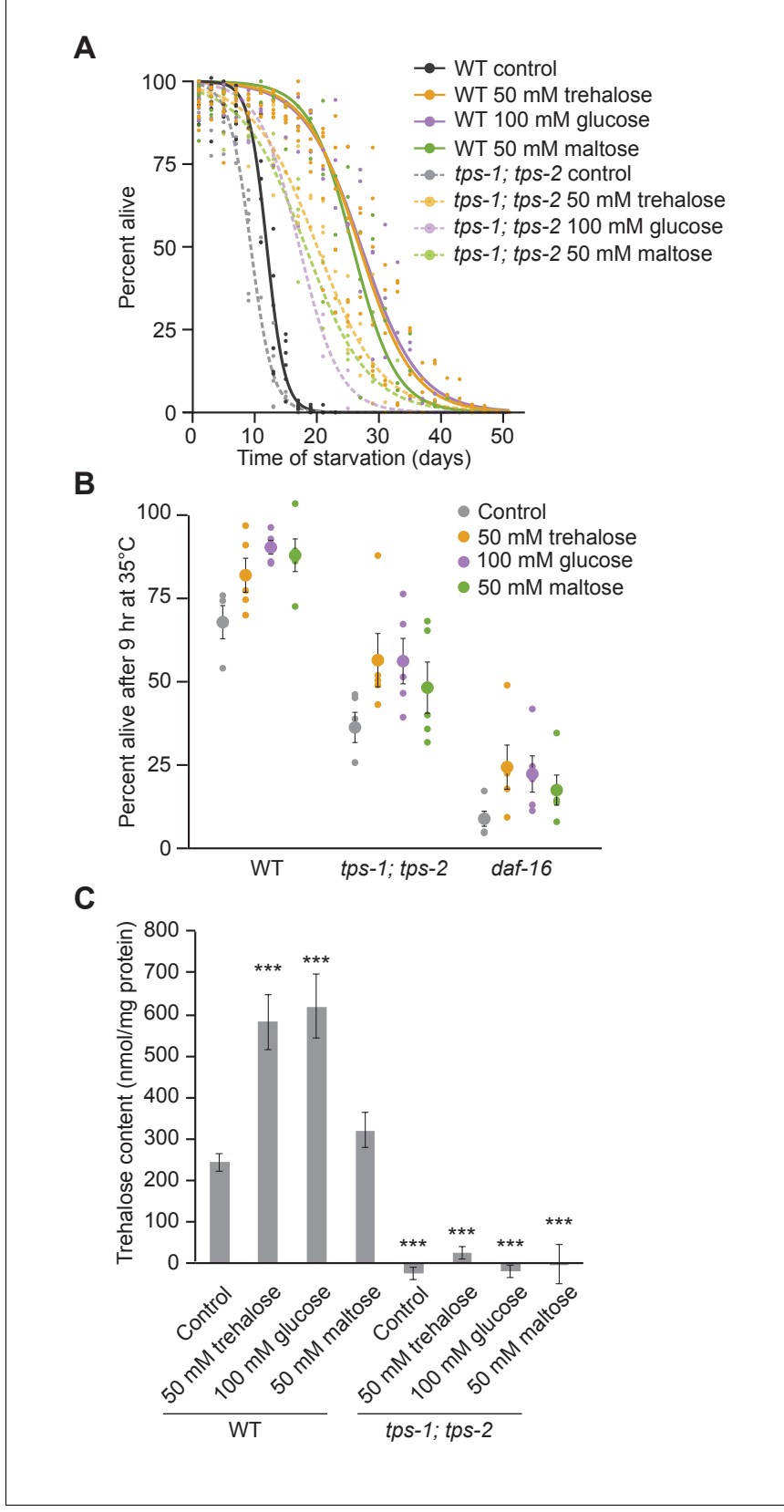

**Figure 6.** Interconversion of trehalose and glucose promotes resistance to starvation and heat. (A) Logistic regression survival curves are plotted for wild-type (WT) and *tps-1; tps-2* supplemented with a variety of sugars. (B)

*Figure 6 continued on next page*

*Figure 6 continued*

Heat shock survival on the first day of L1 starvation is plotted for WT, *tps-1; tps-2* and *daf-16* with and without supplementation with 50 mM trehalose, 100 mM glucose, and 50 mM maltose. (C) Trehalose content is shown for WT and *tps-1; tps-2* supplemented with different sugars on the first day of L1 starvation. With the exception of maltose supplementation, all conditions measured were significantly different from the WT control (p<0.001, unpaired t-tests, n = 7).

DOI: https://doi.org/10.7554/eLife.30057.020

The following source data is available for figure 6:

**Source data 1.** Raw data for display items in *Figure 6*.

DOI: https://doi.org/10.7554/eLife.30057.021

two expression profiles remained distinct (*Figure 7B*, *Figure 7—figure supplement 1*). This incomplete effect of supplementation in *tps-1; tps-2* on gene expression is analogous to the effect of supplementation on survival (*Figure 6A*), and it too suggests that conversion of supplemental trehalose into glucose and back into trehalose is necessary to produce a complete physiological effect.

## Trehalose fuels cell division in permissive conditions

Expression analysis provided an opportunity to identify genes whose expression is affected by the use of trehalose as an energy source during starvation. The effects of trehalose supplementation in *tps-1; tps-2* mutants suggest that supplementary trehalose functions as an energy source rather than a stress protectant in this background (*Figure 6*). We examined the set of genes differentially expressed in *tps-1; tps-2* mutants in response to trehalose supplementation, and they are enriched for GO terms related to DNA synthesis and replication (*Figure 7C*), an energetically expensive process. All eight genes driving this association were upregulated by trehalose supplementation. We hypothesized that trehalose is used as an energy source to fuel DNA synthesis and possibly cell division. We examined two cell lineages that divide during the L1 stage in fed larvae. Hypodermal seam cells of the V lineage divide about 5 hr after hatching with food (*Sulston and Horvitz, 1977*). *daf-16* mutants fail to arrest seam cell divisions during L1 arrest if they are supplemented with ethanol, but penetrance is incomplete and divisions are much slower than in fed larvae (*Baugh and Sternberg, 2006*; *Kaplan et al., 2015*). We potentiated cell division using a *daf-16* mutant without ethanol, and we found that trehalose supplementation promoted seam cell division in these permissive conditions (*Figure 7D*). The M mesoblast cell lineage begins dividing about 9 hr after hatching in fed larvae (*Sulston and Horvitz, 1977*). Like the seam cells, M lineage cell divisions occur in *daf-16* mutants during L1 arrest if supplemented with ethanol (*Baugh and Sternberg, 2006*; *Fukuyama et al., 2015*; *Kaplan et al., 2015*). Trehalose supplementation did not promote M lineage divisions in otherwise starved WT larvae, even with ethanol, but the number of M lineage divisions was increased by α–α trehalose supplementation in *daf-16* mutants (*Figure 7E*, *Figure 7—figure supplement 2*). *daf-16* mutants supplemented with α–α trehalose were also more likely to have multiple M lineage divisions: 16% had two or more divisions (at least four cells), compared to 6% in *daf-16* without supplementation (n = 1200 and 1205 animals, respectively). Furthermore, supplementation with non-hydrolyzable α–β trehalose did not promote M lineage division (*Figure 7E*), confirming that this effect on cell division reflects the use of trehalose as a glycolytic input.

Endogenous trehalose promotes cell division in permissive conditions. Mutation of *tps-1* and *tps-2* in a *daf-16*/FoxO background reduced the number of M-cell divisions with ethanol but no trehalose supplementation (*Figure 7F*, p=0.005, unpaired t-test, n = 4), suggesting endogenous trehalose promotes cell division in permissive conditions. Similar to trehalose supplementation, glucose supplementation promoted M lineage divisions in *daf-16* with ethanol (*Figure 7F*, p=0.04, unpaired t-test, n = 4). However, in *tps-1; tps-2; daf-16* mutants, these sugars had no effect on cell division (*Figure 7F*). This result shows that cell divisions promoted by sugar supplementation require endogenous trehalose synthesis, implying that supplementary sugar must be converted to trehalose to promote cell division.

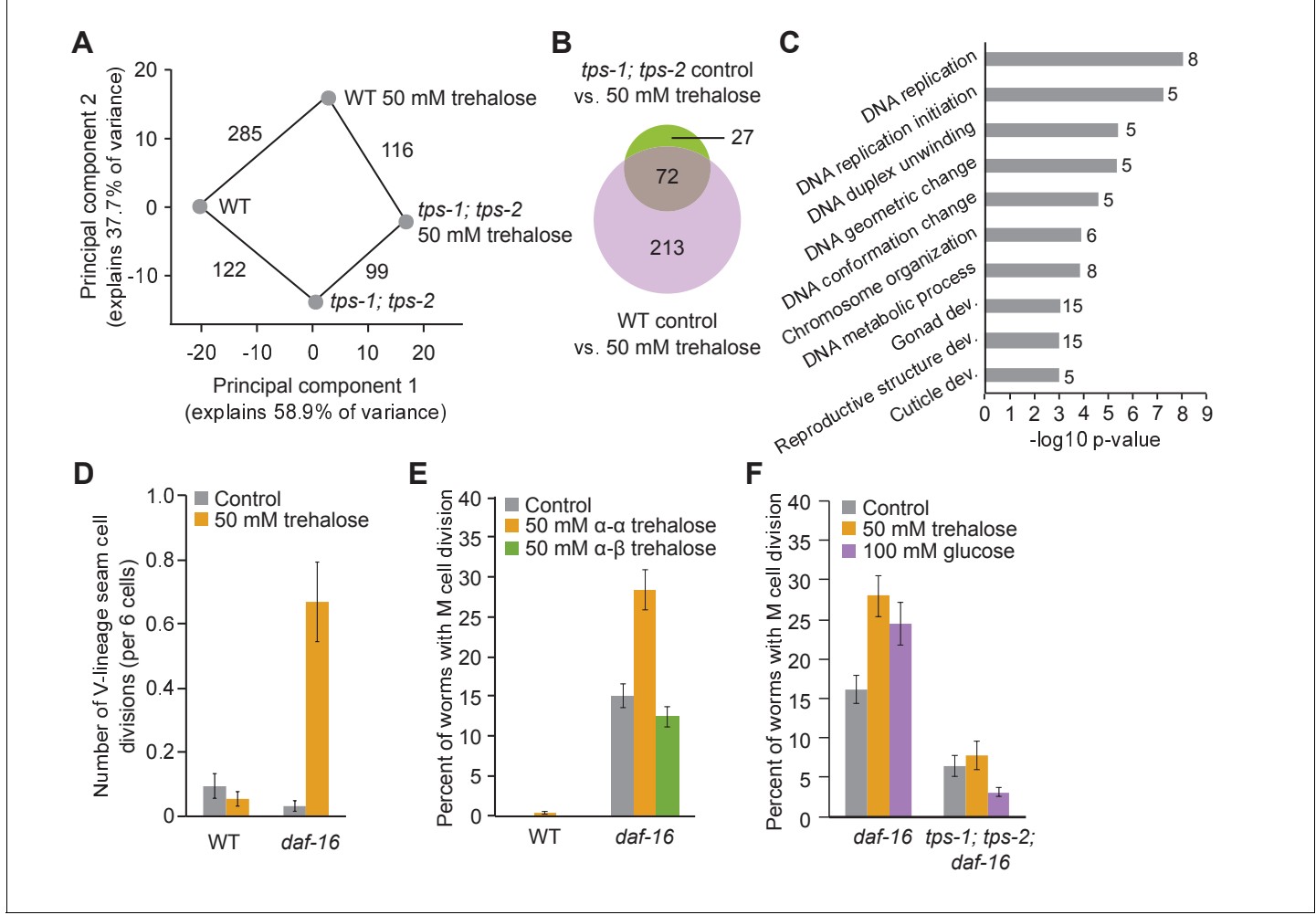

**Figure 7.** Trehalose fuels cell division in permissive conditions. (**A**) Principal component analysis (PCA) of wild-type (WT) and *tps-1; tps-2* mutant gene expression with and without 50 mM trehalose. The number of differentially expressed genes between each condition is indicated on the connecting lines. (**B**) A Venn diagram shows significant overlap between genes changing in response to trehalose in both WT and *tps-1; tps-2* (FDR < 0.05). The genes changing in *tps-1; tps-2* mutants are largely a subset of those that also change in WT. (**C**) –log10 p-values are plotted for GO terms enriched among genes with differential expression between *tps-1; tps-2* and *tps-1; tps-2* with 50 mM trehalose supplementation. The number of genes contributing to each GO term is listed beside each bar. (**D**) The number of hypodermal seam cell divisions per worm after 4 d of L1 starvation is shown for WT and *daf-16*. Mean and standard error of the mean (SEM) of t replicates is plotted. (**E**) The percentage of worms with M-cell divisions after 8 d of L1 starvation is shown for WT and *daf-16* in control buffer (n = 6) and supplemented with 50 mM α–α trehalose (n = 6) or 50 mM α–β trehalose (n = 3). Worms were in S-basal (with ethanol and cholesterol). Mean and SEM are plotted. (**F**) The percentage of worms with M-cell divisions after 8 d of L1 starvation is shown for *daf-16* and *tps-1; tps-2; daf-16* worms supplemented with 100 mM glucose or 50 mM trehalose. *tps-1; tps-2; daf-16* worms have fewer divisions than *daf-16* mutants (p=0.005, unpaired t-test, n = 4). Supplemental trehalose and glucose promote cell division in *daf-16* worms (p=0.01 and 0.04, respectively), but do not significantly affect divisions in *tps-1; tps-2; daf-16* mutants.

DOI: https://doi.org/10.7554/eLife.30057.022

The following source data, source code and figure supplements are available for figure 7:

**Source code 1.** Code used for analysis of RNA-seq experiment.
DOI: https://doi.org/10.7554/eLife.30057.025
**Source data 1.** Raw data for display items in *Figure 7*.
DOI: https://doi.org/10.7554/eLife.30057.026
**Source data 2.** Results of RNA-seq comparing WT and *tps-1; tps-2* mutant worms with and without trehalose supplementation.
DOI: https://doi.org/10.7554/eLife.30057.027
**Figure supplement 1.** Expression analysis of WT and *tps-1; tps-2* with and without trehalose supplementation.
DOI: https://doi.org/10.7554/eLife.30057.023
**Figure supplement 2.** Representative images of M-cell divisions.
DOI: https://doi.org/10.7554/eLife.30057.024

## Discussion

We sought to determine how *daf-16*/FoxO promotes resistance to acute starvation. We found that *daf-16* rapidly activates transcription of metabolic enzymes that shift carbon flux toward the glyoxylate shunt and gluconeogenesis to drive trehalose synthesis (*Figure 8*). These transcriptional changes lead to increased levels of trehalose in the first hours of starvation, and this shift in metabolic flux confers starvation resistance. That is, disruption of the glyoxylate shunt, gluconeogenesis, or trehalose synthesis reduces starvation survival, while supplementation with trehalose increases survival. We provide multiple lines of evidence that trehalose has dual physiological functions to support survival, serving as a stress protectant and an energy input for glycolysis. Active trehalose synthesis is required to maintain a pool of trehalose during starvation, even with trehalose supplementation, suggesting interconversion of trehalose and glucose is necessary for full physiological function.

### *daf-16*/FoxO shifts central carbon metabolism during starvation

We define a set of early targets of *daf-16*/FoxO in the ecologically and developmentally relevant context of L1 starvation (*Baugh, 2013*). We show that enzymes of the glyoxylate shunt, gluconeogenesis, and trehalose synthesis are transcriptionally upregulated during L1 starvation. These findings corroborate expression analysis of dauer larvae, suggesting a conserved starvation response across developmental stages despite unique features of dauer larvae (*Braeckman, 2009*; *Erkut et al., 2016*; *McElwee et al., 2006*; *Wang and Kim, 2003*). Furthermore, we show that changes in expression have physiological consequence, affecting trehalose levels and starvation resistance. We provide evidence that starved larvae are sensitive to disruption of the glyoxylate shunt, which feeds into gluconeogenesis, in contrast to fed larvae. Conversely, starved larvae are relatively insensitive to disruption of glycolysis, in contrast to fed larvae. Thus, metabolic flux shifts from the TCA cycle and electron transport chain to the glyoxylate shunt and from glycolysis to gluconeogenesis during starvation. We show that this metabolic shift culminates in increased levels of trehalose, and that trehalose supports starvation survival. Expression of *icl-1*, the isocitrate lyase/malate synthase enzyme central to the glyoxylate shunt, is upregulated by starvation at other

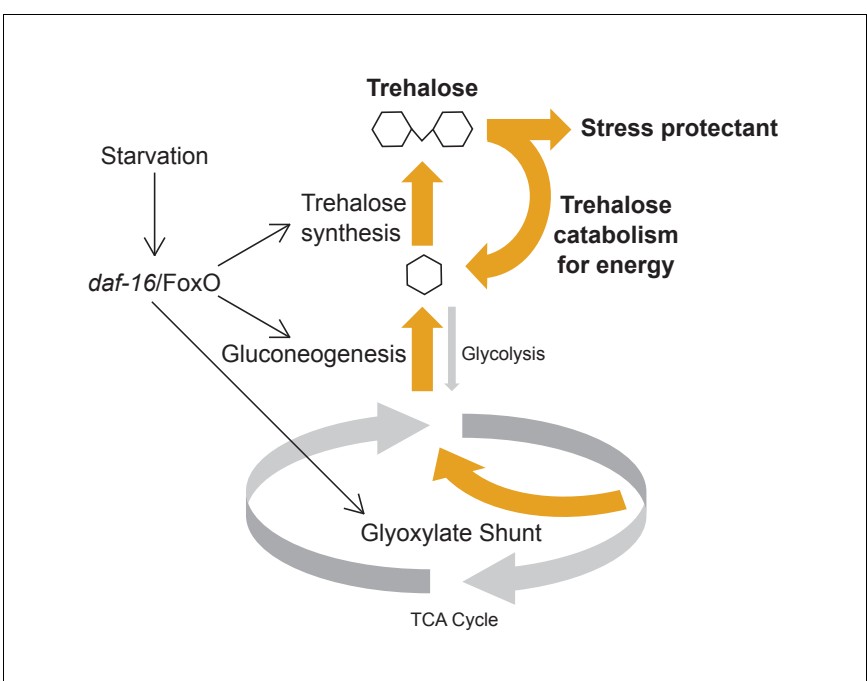

**Figure 8.** Model for metabolic adaptation to acute starvation. *daf-16*/FoxO transcriptionally upregulates enzymes involved in the glyoxylate shunt, gluconeogenesis, and trehalose synthesis to drive an increase in production of trehalose. Trehalose functions as a stress protectant and an energy source to promote survival.
DOI: https://doi.org/10.7554/eLife.30057.028

developmental stages as well (*Liu et al., 1997*), suggesting the metabolic shift we describe is part of a general starvation response rather than being specific to L1 or dauer larvae. Critically, we show that this metabolic shift is under direct regulation of insulin-like signaling via DAF-16/FoxO. That is, DAF-16 binds directly to the *tps-1* promoter and the 14 other metabolic enzymes whose transcription is activated by *daf-16* (*Niu et al., 2011*; *Schuster et al., 2010*; *Zhang et al., 2013*). Mammals do not synthesize trehalose, but mammalian FOXO1 also upregulates gluconeogenesis during nutrient stress, suggesting broad conservation of the effects of insulin-like signaling on metabolic adaptation (*Matsumoto et al., 2007*; *Puigserver et al., 2003*).

Intermediary metabolism is subject to extensive post-translational regulation (*Arif et al., 2017*), and such regulation likely reinforces the patterns of transcriptional regulation we describe for L1 starvation. Although the changes in mRNA abundance we report are statistically significant with concerted effects on carbon metabolism, in many cases the fold-changes are relatively modest. It is likely that these patterns of transcriptional regulation are reinforced by post-translational control, possibly even under the control of insulin-like signaling. Indeed, enzymes involved in glycolysis and gluconeogenesis are known to be subject to post-translational modification (*Tripodi et al., 2015*). Thus, the expression patterns we report are consistent with the changes in metabolic flux observed during starvation, but transcription is unlikely to be the sole cause for these physiological changes. The integration of transcriptional and post-translational regulation of metabolism during starvation is an exciting area for further research.

We show that trehalose synthesis is an effector mechanism by which reduced insulin-like signaling and *daf-16*/FoxO activity promote starvation resistance. Disruption of trehalose synthesis reduces starvation survival, and supplementation of otherwise starved worms with trehalose increases survival. We also show that trehalose synthesis is required for heat resistance during L1 starvation, and that supplemental trehalose increases resistance. Extended starvation survival of a *daf-2*/InsR mutant depends on trehalose synthesis, indicating that *daf-16*/FoxO-dependent induction of the three enzymes that mediate trehalose synthesis (*tps-1*, *tps-2,* and *gob-1*) contributes to starvation resistance and likely cross-tolerance to other stressors such as heat, high salt, and freezing (*Baugh, 2013*). However, disruption of trehalose synthesis does not completely eliminate increased survival of *daf-2* mutants. Likewise, trehalose supplementation of a *daf-16* mutant increases survival, but not to the level of WT worms supplemented with trehalose. We believe supplementation of *daf-16* mutants with trehalose does not have more of an effect on survival because induction of *tps-1* and *tps-2* is abrogated by disruption of *daf-16*, and trehalose synthesis is necessary for the full physiological effect of supplementation. Furthermore, these results imply that trehalose synthesis is not the only effector of reduced insulin-like signaling and DAF-16 activation, suggesting functional contribution of other DAF-16 targets to starvation resistance. Nonetheless, trehalose synthesis is a potent target of DAF-16 with substantial effects on starvation physiology.

## Dual function of trehalose to support starvation survival

Genetic, pharmacological, and biochemical approaches reveal dual function of trehalose as a glycolytic input and stress protectant. The glyoxylate shunt and trehalose synthesis are required for desiccation tolerance in dauer larvae (*Erkut et al., 2011*; *Erkut et al., 2016*). Likewise, mutants with reduced insulin-like signaling are resistant to osmotic stress, and disruption of *tps-1* and *tps-2* suppresses this effect (*Lamitina and Strange, 2005*). In addition, trehalose supplementation improves viability during cryopreservation of *C. elegans* (Kevin O'Connell, personal communication; see Worm Breeder's Gazette). These results suggest that trehalose supports viability in conditions where water is limiting, as in other organisms. Trehalose functions as such a stress protectant by preserving membrane organization and protein structure (*Erkut et al., 2011*; *2012*; *Leekumjorn and Sum, 2008*). Supplementation of starved L1 larvae with non-hydrolyzable α–β trehalose increases survival, suggesting similar function as a stress protectant during starvation, despite ample water. However, supplementation with α–β trehalose does not increase survival to the extent that hydrolyzable α–α trehalose does, suggesting catabolism of trehalose and use as a glycolytic input also contributes to starvation survival. Likewise, trehalose supplementation in worms with all five trehalase genes mutated extends survival, providing additional evidence of trehalose function as a stress protectant. However, extension of survival is limited in the quintuple mutant compared to WT, further showing that catabolism of trehalose is necessary for its full effect. In addition, pharmacological inhibition of glycolysis limits the increase in survival provided with trehalose supplementation in WT, consistent

with the glucose produced by catabolism being used for glycolysis. By uncoupling the effects of trehalose, these results provide multiple lines of evidence that supplemental trehalose supports starvation survival by functioning as an input for glycolysis as well as a stress protectant. These results also indicate that starvation survival requires maintenance of molecular and cellular integrity as well as adequate energy. Furthermore, analysis of the trehalose synthase double mutant and trehalase quintuple mutant show that cycling of glucose and trehalose is necessary for the dual function and complete physiological effects of trehalose.

The relative physiological significance of endogenous trehalose as an energy source and a stress protectant is difficult to discern. The trehalose synthase double mutant, which lacks trehalose entirely, is starvation sensitive, but it is unclear if its lack of function as a stress protectant, an energy source, or both is responsible. In contrast, the trehalase quintuple mutant, which maintains abnormally high levels of trehalose (higher than those observed in WT with supplementation) but cannot catabolize it for energy, survives starvation similar to WT. This result suggests that abnormally high levels of trehalose and the inability to catabolize it offset each other in their effects on survival. We did observe a reduction in endogenous trehalose levels relative to protein content during long-term starvation, consistent with depletion of it as an energy source. Furthermore, endogenous trehalose (inferred from the *tps-1; tps-2* mutant) promotes cell division in a permissive background, presumably reflecting function as an energy source. However, disruption of glycolysis had a relatively minor effect on starvation survival. Notably, we assessed survival in laboratory conditions, without examination of other organismal properties, like movement or foraging behavior, that could be supported by glycolysis and contribute to survival in the wild.

## A paradox of sugar synthesis during starvation

It is somewhat surprising that gluconeogenesis is upregulated during starvation. One might naively imagine that sugars are catabolized for energy during starvation and that fat and other nutrient stores are more directly utilized for energy than by fueling synthesis of sugar. That is, a cycle of sugar synthesis and catabolism during starvation is seemingly futile. Function of trehalose as a stress protectant during starvation provides some resolution to this paradox, but upregulation of gluconeogenesis during starvation also occurs in mammals (*Puigserver et al., 2003*), though they do not produce trehalose. In humans, fasting and starvation induce hepatic gluconeogenesis (*Rothman et al., 1991*), and dysregulation of hepatic glucose production is a hallmark of diabetes (*Consoli, 1992*; *Lin and Accili, 2011*; *Titchenell et al., 2017*). Glucose produced in the liver is circulated to other organs with high-energy demand such as skeletal muscle and the brain. Thus, in mammals, glucose transport between organs resolves the paradox of sugar synthesis during starvation.

We speculate that in nematodes trehalose is synthesized in the hypodermis and intestine during starvation and transported to other organs to satisfy energy demand. Indeed, insects transport trehalose instead of glucose: trehalose synthesized in the fat body is circulated through the hemolymph to fuel flight and other metabolic processes (*Candy and Kilby, 1961*; *Clegg and Evans, 1961*; *Saito, 1963*; *Thompson, 2003*; *Treherne, 1958*; *Wyatt and Kale, 1957*). Given the apparent absence of a glucose-6-phosphatase enzyme in *C. elegans*, there has been speculation that trehalose is the primary transport sugar (*McElwee et al., 2006*). We observe expression of *tps-1* and *tps-2* in the intestine and hypodermis, which are also fat depots and primary sites of energy storage (*Mullaney and Ashrafi, 2009*). The phosphatase *gob-1*, which converts trehalose 6-phosphate to trehalose, is also expressed in the intestine (*Kormish and McGhee, 2005*). In addition, catabolism of trehalose in neurons may support glycolysis and neuronal function during starvation (*Jang et al., 2016*), which may be required for appropriate foraging behavior and adaptation to starvation in the wild (*Kniazeva et al., 2015*). In summary, we propose conservation of an integrated organismal response to starvation among nematodes, insects, and mammals that is defined by FoxO transcription factors promoting carbon flux from fatty acids through gluconeogenesis to produce sugar that is transported throughout the animal to support glycolysis.

# Materials and methods

## Key resources table

| Reagent type (species) or resource | Designation | Source or reference | Identifiers | Additional information |
|---|---|---|---|---|
| strain, strain background (*C. elegans*) | N2 (Bristol) | Caenorhabditis Genetics Center | N2 | |
| strain, strain background (*C. elegans*) | *daf-16(mgDf50)* | Caenorhabditis Genetics Center | RRID:WB-STRAIN: GR1307 | |
| strain, strain background (*C. elegans*) | *daf-16(mgDf47)* | Caenorhabditis Genetics Center | PS5150 | |
| strain, strain background (*C. elegans*) | *dpy-5(e907); sEx14885* | Caenorhabditis Genetics Center | RRID:WB-STRAIN: BC14885 | |
| strain, strain background (*C. elegans*) | *dpy-5(e907); sEx14876* | Caenorhabditis Genetics Center | RRID:WB-STRAIN: BC14876 | |
| strain, strain background (*C. elegans*) | *ayIs7[Phlh-8::GFP]* | Caenorhabditis Genetics Center | RRID:WB-STRAIN: PD4667 | |
| strain, strain background (*C. elegans*) | *daf-16(mgDf50); ayIs7[Phlh-8::GFP]* | As described in **Baugh and Sternberg (2006)** DOI: 10.1016/j.cub.2006.03.021, **Kaplan et al. (2015)** DOI: 10.1371/journal.pgen.1005731 | LRB101 | |
| strain, strain background (*C. elegans*) | *icl-1(ok531)* | Caenorhabditis Genetics Center | RRID:WB-STRAIN: RB766 | |
| strain, strain background (*C. elegans*) | *pck-1(ok2098)* | Caenorhabditis Genetics Center | RRID:WB-STRAIN: RB1688 | |
| strain, strain background (*C. elegans*) | *pyk-1(ok1754)* | Caenorhabditis Genetics Center | RRID:WB-STRAIN: VC1265 | |
| strain, strain background (*C. elegans*) | *tps-1(ok373)* | Caenorhabditis Genetics Center | RRID:WB-STRAIN: VC225 | |
| strain, strain background (*C. elegans*) | *tps-2(ok526)* | Caenorhabditis Genetics Center | RRID:WB-STRAIN :RB760 | |
| strain, strain background (*C. elegans*) | *tps-1(ok373); tps-2(ok526)* | **Erkut et al. (2011)** DOI: 10.1016/j.cub.2011.06.064 | | Shared by Teymuras Kurzchalia |
| strain, strain background (*C. elegans*) | *tps-1(ok373); tps-2 (ok526); daf-2(e1370)* | **Erkut et al. (2011)** DOI: 10.1016/j.cub.2011.06.065 | | Shared by Teymuras Kurzchalia |
| strain, strain background (*C. elegans*) | *tre-1(ok327)* | Caenorhabditis Genetics Center | RRID:WB-STRAIN: RB728 | |
| strain, strain background (*C. elegans*) | *tre-2(ok575)* | Caenorhabditis Genetics Center | RRID:WB-STRAIN: RB789 | |
| strain, strain background (*C. elegans*) | *tre-3(ok394)* | Caenorhabditis Genetics Center | RRID:WB-STRAIN: RB1400 | |
| strain, strain background (*C. elegans*) | *tre-4(gk298765)* | Caenorhabditis Genetics Center | LRB327 | 2x backcrossed VC40057 |
| strain, strain background (*C. elegans*) | *tre-5(ok612)* | Caenorhabditis Genetics Center | RRID:WB-STRAIN: RB806 | |
| strain, strain background (*C. elegans*) | *tre-1(ok327); tre-5(ok612); tre-2(ok575); tre-3(ok394); tre-4(gk298765)* | this paper | LRB334 | This strain was generated by crosses to combine each of the individual trehalase mutants listed above. |
| strain, strain background (*C. elegans*) | *dpy-5(e907); sEx14863* | Caenorhabditis Genetics Center | RRID:WB-STRAIN: BC14863 | |
| strain, strain background (*C. elegans*) | *dpy-5(e907); sIs11667* | Caenorhabditis Genetics Center | RRID:WB-STRAIN: BC12475 | |
| strain, strain background (*C. elegans*) | *dpy-5(e907); sEx15383* | Caenorhabditis Genetics Center | RRID:WB-STRAIN: BC15383 | |

*Continued on next page*

*Continued*

| Reagent type (species) or resource | Designation | Source or reference | Identifiers | Additional information |
|---|---|---|---|---|
| strain, strain background (*C. elegans*) | *dpy-5(e907); sEx14865* | Caenorhabditis Genetics Center | RRID:WB-STRAIN: BC14865 | |
| strain, strain background (*C. elegans*) | *syIs78 [Pajm-1::AJM-1::GFP; unc-119+]* | Caenorhabditis Genetics Center | RRID:WB-STRAIN: PS4657 | |
| strain, strain background (*C. elegans*) | *daf-16(mgDf50); syIs78[Pajm-1::AJM-1::GFP]* | As described in **Baugh and Sternberg (2006)** DOI: 10.1016/j.cub.2006.03.021, **Kaplan et al. (2015)** DOI: 10.1371/journal.pgen.1005731 | LRB240 | |
| strain, strain background (*C. elegans*) | *tps-1(ok373); tps-2(ok526); daf-16(mgDf50); syIs78 [Pajm-1::AJM-1::GFP]* | this paper | | Strain LRB240 was crossed into *tps-1(ok373); tps-2(ok526)* |
| commercial assay or kit | *C. elegans* expression array | Affymetrix | 900383 | |
| commercial assay or kit | NEBNext Ultra RNA Library Prep Kit for Illumina | New England BioLabs | E7530 | |
| commercial assay or kit | Bicinchoninic acid kit | Sigma | BCA1 | |
| commercial assay or kit | Trehalose assay kit | Megazyme | K-TREH | |
| commercial assay or kit | Qubit protein assay kit | Thermo Fisher Scientific | Q33211 | |
| chemical compound, drug | 2-deoxy-D-glucose | Sigma | 154-17-6 | |
| chemical compound, drug | 3-mercaptopicolinic acid (3 MPA) | Santa Cruz Biotechnology | 320386-54-7 | |
| chemical compound, drug | D-(+)-Trehalose dihydrate | Sigma | 6138-23-4 | |
| chemical compound, drug | α,β-trehalose | Omicron Biochemicals Inc. | TRE-006 | |
| software, algorithm | WormSizer | **Moore et al. (2013)** DOI: 10.1371/journal.pone.0057142 | | https://github.com/ bradtmoore/wormsizer |

## Nematode culture and strains

The N2 Bristol strain was used and is referred to as WT. Worm stocks were maintained according to standard culture methods with nematode growth medium (NGM) and OP50 *E. coli*. Strains used include: GR1307 *daf-16(mgDf50)*, PS5150 *daf-16(mgDf47)*, BC14885 *dpy-5(e907); sEx14885*, BC14876 *dpy-5(e907); sEx14876*, PD4667 *ayIs7[Phlh-8::GFP]*, LRB101 *daf-16(mgDf50); ayIs7[Phlh-8:: GFP]*, RB766 *icl-1(ok531)*, RB1688 *pck-1(ok2098)*, VC1265 *pyk-1(ok1754)*, *tps-1(ok373)*, *tps-2(ok526)*, *tps-1(ok373); tps-2(ok526)*, *tps-1(ok373); tps-2(ok526); daf-2(e1370)*, RB728 *tre-1(ok327)*, RB789 *tre-2(ok575)*, RB1400 *tre-3(ok394)*, LRB327 *tre-4(gk298765)* (2x backcrossed VC40057), RB806 *tre-5 (ok612)*, LRB334 *tre-1(ok327); tre-5(ok612); tre-2(ok575); tre-3(ok394); tre-4(gk298765)*, BC14863 *dpy-5(e907); sEx14863*, BC12475 *dpy-5(e907); sIs11667*, BC15383 *dpy-5(e907); sEx15383*, BC14865 *dpy-5(e907); sEx14865*, PS4657 *syIs78[Pajm-1::AJM-1::GFP and unc-119+]*, LRB240 *daf-16(mgDf50); syIs78[Pajm-1::AJM-1::GFP]*, *tps-1(ok373); tps-2(ok526); daf-16(mgDf50); ayIs7[Phlh-8::GFP]*.

## Microarray

Embryos of WT (N2) and GR1307 *daf-16(mgDf50)* worms were subjected to a double-bleach procedure to isolate staged embryos as described elsewhere (*Baugh, 2009*; *Baugh et al., 2009*). Embryos were cultured in S-basal for 16 hr at 20°C, washed with S-basal by centrifugation and frozen in liquid nitrogen. At this collection time, all embryos had hatched and L1 larvae had been arrested for 2–4 hr. RNA was isolated using TRIzol (Invitrogen, Carlsbad, CA) according to the manufacturer's specifications. RNA was further purified using RNeasy micro kit (Qiagen, Valencia, CA). Spectrophotometry was used to assess RNA purity and concentration, and gel electrophoresis was used to confirm its integrity. 100 ng of total RNA was amplified and labeled as cRNA using the MessageAmp II-Biotin Enhanced kit (Ambion, Foster City, CA) according to the manufacturer's protocol. Molecular weight

and concentration of the biotin-labeled cRNA product was assessed with an Agilent BioAnalyzer and spectrophotometer. 12.5 µg of cRNA was fragmented, hybridized to the *C. elegans* expression array (Affymetrix, Santa Clara, CA), and scanned according to the manufacturer's instructions.

Microarray data were analyzed using the LIMMA package in R. Each probe set was mapped to WS180 as in (*Baugh et al., 2009*). Normalized log2 GCRMA values were used to assess significance of expression changes with the LIMMA/GCRMA empirical Bayes test. Genes significantly different between conditions were hierarchically clustered with Euclidian distances using Gene Cluster 3.0 (*Figure 1—figure supplement 1*). This same group of genes was used for principal component analysis (*Figure 1A*).

## Metabolomics

To measure metabolites in WT and *daf-16*/FoxO mutant worms, both strains were grown on 10 cm NGM plates with OP50 *E. coli*. These populations were bleached to isolate embryos. Embryos were arrested in S-complete and were either maintained in S-complete (starvation) or had 25 mg/mL HB101 *E. coli* added to culture upon hatching (fed). Worms were washed in S-basal and flash frozen in 0.6% formic acid. Worms were lysed by sonication with a Bioruptor at 4°C for 15 min with 30 s on, 30 s off at high power. Sonicated samples were stored at −80°C.

To extract metabolites, all samples were thawed on ice and then sonicated on ice, using ten 30 s on–off pulses at 30% power using a Series 60 Sonic Dismembrator (Model F60, Fisher Scientific). An aliquot was saved for total protein analysis via bicinchoninic acid assay (BCA Sigma, St. Louis, MO). Remaining sample was mixed with acetonitrile (1:1 ratio), vortexed, and divided into five parts. Three parts of the sample was used for the quantification of amino acid and acyl carnitine, one part for organic acids, and from the remaining one part sample equivalent to 500 µg of protein was used for non-targeted metabolomics analysis. Amino acids and acylcarnitines were analyzed using MS/MS as previously described (*An et al., 2004*; *Wu et al., 2004*) and organic acids were quantified using GC/MS as described in the study by *Jensen et al. (2006)*. Non-targeted metabolomics analysis was performed using GC/MS metabolomics as described in the study by *Banerjee et al. (2015)*: Raw data were normalized to protein levels, and converted to log2 fold change values relative to fed WT worms.

## Starvation survival

Embryos were isolated by standard hypochlorite treatment and arrested at a density of 1/µL in virgin S-basal (no ethanol added) in glass test tubes. Tubes were kept on a roller drum at 21–22°C. For survival experiments with 2-DG, 3-mercaptopicolinic acid, trehalose, glucose, and maltose, these compounds were added to virgin S-basal before addition of embryos. To score survival, 100 µL samples of each culture were plated next to a lawn of OP50 *E. coli* on 6 cm NGM plates. The total number of worms plated ($T_p$) was counted after plating and 2 d later the number of worms alive and able to move to the bacterial lawn ($T_a$) was scored. Survival was calculated as $T_a/T_p$, logistic survival curves were fit to the data, and median survival time was calculated (*Artyukhin et al., 2013*; *Hibshman et al., 2016*; *Kaplan et al., 2015*).

## Trehalose measurement

We used the Megazyme Trehalose Assay Kit (K-Treh) to measure the levels of trehalose in worms. Briefly, washed worms were flash frozen in 100 µL of 0.6% formic acid. Worms were sonicated for lysis, as with metabolomics analysis. Samples were split for protein analysis and trehalose analysis. Trehalose was measured according to the protocol provided with the Megazyme kit. Absorbances were measured on a Nanodrop. Protein concentration was quantified with a Qubit protein assay kit (Thermo Fisher Scientific, Waltham, MA). Levels of trehalose were normalized to protein. Normalizing trehalose content by worm number instead of protein concentration provided qualitatively similar results.

## Worm size measurements

The Wormsizer plugin for FIJI was used to measure the length of worms (*Moore et al., 2013*). Length after 48 hr of development was measured in several genotypes and in the presence of a range of 2-DG and 3-mercaptopicolinic acid concentrations (*Figure 3B–D*). Length of starved L1

larvae was similarly measured (*Figure 4K*). Worms were washed in S-basal, plated on unseeded 10 cm NGM plates and imaged on a Zeiss Discovery.V20 stereomicroscope. Images were processed with Wormsizer.

## Fluorescent reporter analysis

To quantify fluorescence of P*tps-1*::GFP and P*tps-2*::GFP strains (*Figure 4B–G*) images were acquired at 40x with consistent exposure times (400 ms) on a Zeiss Imager.A1 outfitted with an Axiocam 506 Mono. Analysis was conducted in Fiji by outlining worms and calculating the average pixel intensity per worm. Background was subtracted from these measurements. *Figure 4D,G* show the distribution of relative intensities of individual worms from three independent biological replicates. P-values were calculated from unpaired t-tests on the mean fluorescent intensities from three biological replicates. Reporter strains were observed at 100x to determine sites of expression.

## Heat shock

Embryos were obtained through standard hypochlorite treatment and arrested in 16 mm glass tubes with virgin S-basal buffer and virgin S-basal with addition of various sugars. L1 worms in liquid culture were kept at 35°C for 9 hr with shaking at 180 rpm. As with starvation survival, 100 μL of aliquots were plated onto 6 cm plates with NGM and OP50. The total number of worms plated ($T_p$) was counted. After 2 d the total number of worms alive ($T_a$) was scored. Survival was calculated as $T_a/T_p$.

## RNA-seq

N2 and *tps-1; tps-2* worms were cultured on standard NGM plates seeded with OP50. Cultures were bleached to isolate embryos, which were suspended 1/μL in virgin S-basal or virgin S-basal with 50 mM trehalose. Samples were flash frozen on liquid nitrogen 24 hr after bleach. RNA was extracted with trizol and chloroform. Libraries were prepared using the NEBNext Ultra RNA Library Prep Kit for Illumina (E7530) with 500 ng of RNA per library and 12 cycles of polymerase chain reaction. Libraries were sequenced using Illumina HiSeq4000, acquiring single end 50 bp reads. Bowtie was used to map reads (*Langmead et al., 2009*). EdgeR was used to assess differential expression from count tables (*Robinson et al., 2010*). An ANOVA-like test in EdgeR identified 750 genes with variability across conditions (FDR < 0.05). These were hierarchically clustered with Euclidian distances using Gene Cluster 3.0 (*Figure 7—figure supplement 1*). Pairwise comparisons between conditions were calculated with the edgeR exact test. Genes different in any pairwise comparison were included in CA (*Figure 6C*).

## GO term analysis

The GOrilla gene ontology enrichment analysis and visualization tool was utilized to determine significant enrichment of processes, functions, and components (*Eden et al., 2009*). In each case, a list of differentially expressed genes was tested against the background set of detected genes.

## Cell division analysis

The PS4657 and LRB240 strains were used to visualize seam cells. The number of cell divisions after 4 d of starvation in virgin S-basal was scored on a Zeiss Imager.A1 compound microscope. The *ayIs7* [P*hlh-8*::GFP] transgene was used to visualize M cells in different genetic backgrounds, and divisions were scored after 8 d of starvation in S-basal buffer (with ethanol and cholesterol added).

## Statistical analysis

Statistics were calculated based on independent biological replicates performed on different days with all experimental methods performed independently of other replicates. Statistical analysis was performed in Microsoft Excel and R Studio. Statistical tests and the number of replicates are described when p-values are reported. Throughout the figures a single asterisk indicates $p < 0.05$, double asterisks indicates $p < 0.01$, and triple asterisks indicate $p < 0.001$.

## Acknowledgements

We thank Olga Ilkayeva, James R Bain, and Michael J Muehlbauer of the Duke Molecular Physiology Institute Metabolomics Lab. The laboratory of Teymuras Kurzchalia provided us with the *tps-1; tps-2* and *tps-1; tps-2; daf-2* strains. Some strains were provided by the CGC, which is funded by NIH Office of Research Infrastructure Programs (P40OD010440). This work was funded by the National Science Foundation (IOS-1120206, LRB) and the National Institutes of Health (R01GM117408-01, LRB).

## Additional information

### Funding

| Funder | Grant reference number | Author |
|---|---|---|
| National Science Foundation | IOS-1120206 | L Ryan Baugh |
| National Institutes of Health | R01GM117408-01 | L Ryan Baugh |

The funders had no role in study design, data collection and interpretation, or the decision to submit the work for publication.

### Author contributions

Jonathan D Hibshman, Conceptualization, Resources, Data curation, Supervision, Investigation, Writing—original draft, Writing—review and editing; Alexander E Doan, Anthony Hung, Dhaval P Bhatt, Investigation, Writing—review and editing; Brad T Moore, Amy K Webster, Data curation, Formal analysis, Writing—review and editing; Rebecca EW Kaplan, Resources, Investigation, Writing—review and editing; Rojin Chitrakar, Investigation; Matthew D Hirschey, Resources, Supervision, Writing—review and editing; L Ryan Baugh, Conceptualization, Resources, Data curation, Supervision, Funding acquisition, Investigation, Writing—original draft, Writing—review and editing

### Author ORCIDs

Jonathan D Hibshman http://orcid.org/0000-0001-7324-9382
Rebecca EW Kaplan https://orcid.org/0000-0002-6237-1435
Anthony Hung http://orcid.org/0000-0002-4725-0635
Amy K Webster https://orcid.org/0000-0003-4302-8102
L Ryan Baugh http://orcid.org/0000-0003-2148-5492

### Decision letter and Author response

Decision letter https://doi.org/10.7554/eLife.30057.037
Author response https://doi.org/10.7554/eLife.30057.038

## Additional files

### Supplementary files

• Supplementary file 1. Comprehensive table of starvation survival statistics.
DOI: https://doi.org/10.7554/eLife.30057.030
• Transparent reporting form
DOI: https://doi.org/10.7554/eLife.30057.031

### Major datasets

The following datasets were generated:

| Author(s) | Year | Dataset title | Dataset URL | Database, license, and accessibility information |
|---|---|---|---|---|
| Baugh LR, Moore BT, Hibshman JD | 2017 | Wild-type and daf-16(mgDf50) L1 larvae 3 hours after hatching in the presence or absence of food | https://www.ncbi.nlm.nih.gov/geo/query/acc.cgi?acc=GSE99201 | Publicly available at the NCBI Gene Expression Omnibus (accession no: GSE99201) |
| Webster AK, Hibshman JD, Chitrakar R, Baugh LR | 2017 | mRNA-seq comparison of N2 and tps1;tps2 double mutants with and without trehalose supplementation | https://www.ncbi.nlm.nih.gov/geo/query/acc.cgi?token=erabokmev-bepxoz&acc=GSE98544 | Publicly available at the NCBI Gene Expression Omnibus (accession no: GSE98544) |

The following previously published dataset was used:

| Author(s) | Year | Dataset title | Dataset URL | Database, license, and accessibility information |
|---|---|---|---|---|
| Baugh LR, Sternberg PW | 2009 | Temporal expression analysis of C. elegans larvae hatching in the presence and absence of food. | https://www.ncbi.nlm.nih.gov/geo/query/acc.cgi?acc=GSE11055 | Publicly available at the NCBI Gene Expression Omnibus (accession no: GSE11055) |

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
