## [Decision Letter]

Thank you for submitting your article "*daf-16*/FoxO promotes gluconeogenesis and trehalose synthesis during starvation to support survival" for consideration by *eLife*. Your article has been reviewed by three peer reviewers, and the evaluation has been overseen by a Guest Reviewing Editor and Mark McCarthy as the Senior Editor. The reviewers have opted to remain anonymous.

The reviewers have discussed the reviews with one another and the Reviewing Editor has drafted this decision to help you prepare a revised submission.

The work is of high quality and important to the field, but the authors should put their valuable data in a broader perspective, as suggested by the reviewers. They should carry out the following actions, and address all remaining points raised by the reviewers, though further experimentation will not be necessary to prove their main claims.

• Comparison of their starvation model with the already established dauer starvation physiology and with mammalian physiology, highlighting what is new in their findings.

• Provide a more precise definition of the 'macromolecular buffer' concept or switch to a more appropriate terminology throughout the manuscript.

• Discuss the role of post-translational modifications in the metabolic changes that may support starvation survival.

• Shorten the discussion on trehalose transport among tissues as no direct evidence is presented in the manuscript. However, the authors may want to refer in that context to the PhD of Y. Feng (University of Bath), not published in a regular journal, but publicly available (http://opus.bath.ac.uk/25409/). They have characterized the *C. elegans* glucose/trehalose transporter.

*Reviewer #1:*

Manuscript by Hibshman et al. presents data that DAF-16 promotes gluconeogenesis and trehalose biosynthesis during L1 arrest. This is based on a comparison of transcriptional activities in WT or *daf-16* mutants that are fed or starved. In addition, metabolomic analysis under these conditions is conducted. According to authors, synthesized trehalose plays dual function: As a source of glucose for glycolysis and as macromolecular buffer (see discussion later). Experimental part of the manuscript is of a high standard. I have, however, a strong concern about two conceptual issues that build the major core of the manuscript.

1) According to definition, starvation is a severe deficiency in caloric intake needed to maintain an organism's life. Thus, an organism in order to survive starvation should mobilize internal energy sources. It must be mentioned that *C. elegans* delivered already an excellent model for starvation response: Dauer larva. Because they do not feed, dauers are "starving" and use internal lipid deposits for survival. It is well established that dauer larva rearrange their metabolism so that triacylglycerols are used as source for gluconeogenesis that finally produces trehalose. Hereby, the major hub, diverting isocitrate from TCA cycle to malate and then to oxaloacetate in cytosol, is the glyoxylate shunt. It is also shown that dauer larva decrease OXPHOS and oxygen consumption, reducing it to 20% of growing L3 larva. In this manuscript, the authors investigate a starvation response of another model for starvation: Arrested L1 larva. It appears that latter is very similar to that of the dauer. On itself, a comparison of these responses could be interesting. It would be good, however, if authors state clearly that this is a comparative study and do not present transition of L1 metabolism to gluconeogenesis and the involvement of glyoxylate shunt hereby as a big surprise.

2) Another issue is the function of trehalose. The authors state that trehalose has a dual role: It is a macromolecular buffer and secondly an energy source. First of all, what is a macromolecular buffer? If authors introduce a novel notion, then they should define it more precisely. Is it the requirement of trehalose for osmotic or desiccation tolerance? I think the assignment of two functions to trehalose is rather trivial: It is only an issue of concentration. Obviously, during starvation trehalose is used as a source for glycolysis and here low/basal amounts suffice. In order to provide resistance to osmotic or desiccation stress, trehalose for instance in dauers is produced in amounts that are several (about 4 to 5) fold higher than the basal level. Because L1 is not resistant either to osmotic or desiccation stress, trehalose in this response might have obviously only an energetic role.

*Reviewer #2:*

This manuscript by Hibshman et al. approaches the unsolved mystery of how the daf-16/FOXO transcription factor promotes starvation resistance. It builds on the solid foundation of work by the senior author, Ryan Baugh. The conclusions of the paper are that gluconeogenesis and trehalose synthesis are regulated in a *daf-16* dependent manner during starvation and that trehalose serves a dual role in starvation survival, as a fuel and as a biochemical starvation defense. The findings are novel, and lend new insight to the biology of starvation defenses. Given the importance of trehalose in insect species, one major criticism of the novelty of the findings is that of course it is well known that gluconeogenesis is critical for starvation defense in mammals, and further that FoxO is well known to regulate gluconeogenesis and transcription of genes encoding gluconeogenic enzymes in liver together with PGC-1alpha (as the authors note in the Discussion). Without evidence for a conserved role of trehalose across phylogeny to higher eukaryotes, this dampens enthusiasm slightly.

Overall, the work is very rigorous and thorough. I similarly appreciate the use of pharmacologic and complementary genetic approaches. The measurement of corporal trehalose concentrations in supplemented worms makes the point very strongly that trehalose is the responsible moiety.

1) Why does the number of genes differ between Figure 1*daf-16* starved vs. WT starved (650) and Figure 1+245 = 502)? Shouldn't this be the same 650 genes that distinguish both states?

2) Do the authors think that by doing the two-factor analysis to identify their 103 high stringency genes that they might have omitted important genes in the starvation response? For example, if a gene is "on" in both starvation and the fed state, and does not change in WT animals but *daf-16* is necessary for its expression which is important in starvation, would that gene have been unfairly excluded? Asking for significant interactions with condition and genotype may have biased the expression analysis.

3) What time point is used to garner the 3.6 fold increase in trehalose in Figure 2? It seems much more blunted in Figure 2, is this because the change is relative to the same timepoint fed animals or relative to time = 0?

4) While I appreciate the corroboration of genetic with pharmacologic data (Figure 3), is it known that 3-mercaptopicolinic acid inhibits gluconeogenesis in the worm as it does in mammalian hepatocytes?

5) It is very curious that trehalose supplementation reduces starvation-induced decrease in body length. Does trehalose supplementation also alter the rate of loss of protein or loss of triglyceride mass with starvation?

6) The inability to rescue *daf-16* starvation sensitivity (Figure 4) with trehalose, and the greater effect of trehalose in WT animals suggests that a "downstream blockade" prevents *daf-16* mutants from utilizing trehalose effectively. The authors should address this, experimentally if possible. Perhaps forced expression of trehalase will permit more effective use of trehalose?

7) Why is the supplementation effect of 50mm trehalose so much more dramatic in Figure 5 versus Figure 4? Is the effect that noisy? It's almost 2x the effect in Figure 5 versus Figure 4.

8) If trehalose supports starvation survival partially without playing a role as a metabolic fuel, supplementation of the quintuple tre mutant should have the same benefit from a,a trehalose versus a,b trehalose. Similarly, since it is not metabolized, the a,b trehalose supplementation should have the same effect in prolonging starvation survival whether 20mM 2-DG is present or not.

9) The very low levels of trehalose in the trehalose supplemented *tps-1;tps-2* double mutant makes the strong argument that there is some cycling of 6-carbon sugars necessary for starvation survival. I'm not suggesting this experiment, but does inhibition of the tre genes in a *tps-1;tps-2* double mutant restore trehalose levels upon supplementation?

10) More of a comment than a criticism – there is a phenomenon known as the glucose paradox, which shows that liver does not directly metabolize glucose to glycogen. Rather, lactate and pyruvate serve as fuels for glycogen synthesis, and any glucose that serves as a substrate for glycogen synthesis first has to go through three-carbon metabolism. This suggests two pools of glucose in the liver, that which is destined for glycolysis and that which is destined for glycogen synthesis. Is the equivalent true here? For example, are there two pools of glucose, that which is destined for glycolysis and that which is destined for trehalose synthesis? While not entirely analogous, a similar phenomenon could begin to explain the observation that it is not possible to maintain trehalose levels without *tps-1;tps-2*. The fact that glycolysis inhibition doesn't markedly affect starvation survival argues strictly against the likelihood that glucose must pass through three carbon metabolism to be subsequently made into trehalose.

*Reviewer #3:*

Hibsham et al. present a very interesting study of the role of trehalose synthesis during starvation for *C. elegans* survival. This is a well-designed and timely study as trehalose has recently been proposed to enhance cryopreservation in *C. elegans*. The authors use a combination of transcriptomics, metabolomics, and physiological characterization in their study and although there is a significant amount of data presented, several aspects are underdeveloped and do not fully support the claims made.

1) The authors study gluconeogenesis and trehalose synthesis based on their transcriptional profiling studies. The transcripts studies, although significantly different between growth conditions, are not impressive based on magnitude. This is perhaps because most of these enzymes are regulated at the protein level and not transcriptionally. In light of the timeframe of the response it is more likely that PTMs will play an important role. This should be examined, particularly since the glyoxylate shunt, gluconeogenesis and trehalose synthesis are previously documented in starvation and dauer, thus diminishing the novelty of the data presented.

2) The authors state that their findings represent a metabolic "Flux shift", but what is it a flux from? What pathway is being repressed in this case?

3) What is the consequence of increased survival after L1 starvation with/without trehalose? Similarly, does trehalose protect from starvation at other developmental stages?

4) Lastly, the authors present a model for trehalose transport between organs (in the Abstract and Discussion), but there is no evidence of this presented in the data. This is an interesting aspect of this story in light of the organism level phenotypes, which should be supported by the data.

---

## [Author Response]

The work is of high quality and important to the field, but the authors should put their valuable data in a broader perspective, as suggested by the reviewers. They should carry out the following actions, and address all remaining points raised by the reviewers, though further experimentation will not be necessary to prove their main claims.• Comparison of their starvation model with the already established dauer starvation physiology and with mammalian physiology, highlighting what is new in their findings.

We appreciate the opportunity to expand on the context and significance of our results. We have elaborated on our discussion of similarities and differences between dauer larvae and L1 starvation in the Introduction, including reasons why metabolism may not be the same, and setting up our investigation as a comparative analysis. We have also added to the Discussion to suggest broad conservation of the up-regulation of gluconeogenesis by FoxO transcription factors given our results for L1 starvation as well as what was known for dauer larvae and mammals. The key difference between our system and mammalian physiology is the terminal production of trehalose rather than glucose. In both systems, however, it is paradoxical to synthesize sugar during nutrient scarcity. We suggest that this paradox is resolved by transporting glucose/trehalose produced by gluconeogenesis in one organ to other organs to fuel glycolysis.

• Provide a more precise definition of the 'macromolecular buffer' concept or switch to a more appropriate terminology throughout the manuscript.

It was not our intention to introduce a new concept regarding biochemical function of trehalose, and we welcome the opportunity to clarify this. Our intention was to acknowledge the rich body of research showing that trehalose can buffer a variety of organisms from a variety of environmental stress including desiccation, freezing, heat, and osmotic stress. We have revised the Introduction to elaborate on research in this area, describing the association of trehalose with macromolecules to preserve membrane organization and protein structure. We have done our best to use similar language to that used in seminal papers. We have changed our terminology to "stress protectant", which we have explicitly defined in the Introduction (Introduction, fifth paragraph). We also briefly restate what we mean in the appropriate portion of the Results section and again in the Discussion.

• Discuss the role of post-translational modifications in the metabolic changes that may support starvation survival.

We regret overlooking the role of post-translational modifications in our previous submission. We have added an entire paragraph to the Discussion (subsection “*daf-16*/FoxO shifts central carbon metabolism during starvation”, second paragraph) to emphasize that metabolism in general and starvation responses in particular are regulated on many levels, and that post-translational modifications play a central role in maintaining metabolic homeostasis. We state that transcriptional regulation is most likely not the sole source of regulation, and we speculate that the patterns of transcriptional regulation we describe are reinforced by post-translational regulation. We agree that the integration of transcriptional and post-translational regulation of metabolic adaptation to nutrient stress is an important avenue of research but unfortunately beyond the scope of this manuscript.

• Shorten the discussion on trehalose transport among tissues as no direct evidence is presented in the manuscript. However, the authors may want to refer in that context to the PhD of Y. Feng (University of Bath), not published in a regular journal, but publicly available (http://opus.bath.ac.uk/25409/). They have characterized the C. elegans glucose/trehalose transporter.

The reviewers are correct that we have not provided direct support for a trehalose transport model. We have removed mention of transport from the Abstract and other summaries. We have also shortened discussion of trehalose transport in the Discussion and removed it from the sub-heading that mentioned it. In particular, we have removed any statements suggesting we provide direct evidence for this claim. We have kept a brief discussion of trehalose transport at the end of the Discussion, clearly stating it as speculation. We have clarified our reasoning to argue that trehalose transport is a possible solution to the paradox of sugar synthesis during starvation. That is, it is counterintuitive to synthesize trehalose via gluconeogenesis during starvation only to catabolize it via glycolysis. Mammalian liver cells also use gluconeogenesis to produce glucose during starvation, though they do not produce trehalose. Glucose produced by the liver is transported to the brain and muscle to support metabolism, and we suggest something similar for trehalose in insects and nematodes. Though speculative, we believe this strengthens the relationship with mammalian starvation metabolism and offers an explanation for a seemingly futile metabolic cycle. We appreciate the reference to the PhD thesis of Y. Feng, but we elected not to cite it here.

Reviewer #1:Manuscript by Hibshman et al. presents data that DAF-16 promotes gluconeogenesis and trehalose biosynthesis during L1 arrest. This is based on a comparison of transcriptional activities in WT or daf-16 mutants that are fed or starved. In addition, metabolomic analysis under these conditions is conducted. According to authors, synthesized trehalose plays dual function: As a source of glucose for glycolysis and as macromolecular buffer (see discussion later). Experimental part of the manuscript is of a high standard. I have, however, a strong concern about two conceptual issues that build the major core of the manuscript.1) According to definition, starvation is a severe deficiency in caloric intake needed to maintain an organism's life. Thus, an organism in order to survive starvation should mobilize internal energy sources. It must be mentioned that C. elegans delivered already an excellent model for starvation response: Dauer larva. Because they do not feed, dauers are "starving" and use internal lipid deposits for survival. It is well established that dauer larva rearrange their metabolism so that triacylglycerols are used as source for gluconeogenesis that finally produces trehalose. Hereby, the major hub, diverting isocitrate from TCA cycle to malate and then to oxaloacetate in cytosol, is the glyoxylate shunt. It is also shown that dauer larva decrease OXPHOS and oxygen consumption, reducing it to 20% of growing L3 larva. In this manuscript, the authors investigate a starvation response of another model for starvation: Arrested L1 larva. It appears that latter is very similar to that of the dauer. On itself, a comparison of these responses could be interesting. It would be good, however, if authors state clearly that this is a comparative study and do not present transition of L1 metabolism to gluconeogenesis and the involvement of glyoxylate shunt hereby as a big surprise.

We agree that our results are more informative through the comparative lens of dauer metabolism. We have revised the Introduction to compare and contrast L1 starvation and dauer arrest. Specifically, we have added more background and citations of dauer metabolism to the Introduction, and we have pointed out unique features of dauer larvae that have made us hesitant to assume dauer metabolism reflects a universal starvation response. In the Discussion we conclude that metabolic adaptation to L1 starvation is similar to dauer larvae, avoiding any suggestion that the metabolic shift we describe is a big surprise. That said, we do emphasize the physiological significance of that shift revealed through our genetic and pharmacological analyses, which had not been performed in dauer larvae or other whole animals, as well as the critical role of *daf-16*/FoxO in directly regulating that shift.

2) Another issue is the function of trehalose. The authors state that trehalose has a dual role: It is a macromolecular buffer and secondly an energy source. First of all, what is a macromolecular buffer? If authors introduce a novel notion, then they should define it more precisely. Is it the requirement of trehalose for osmotic or desiccation tolerance? I think the assignment of two functions to trehalose is rather trivial: It is only an issue of concentration. Obviously, during starvation trehalose is used as a source for glycolysis and here low/basal amounts suffice. In order to provide resistance to osmotic or desiccation stress, trehalose for instance in dauers is produced in amounts that are several (about 4 to 5) fold higher than the basal level. Because L1 is not resistant either to osmotic or desiccation stress, trehalose in this response might have obviously only an energetic role.

As stated above, we have revised the Introduction to better describe the biochemical mechanism of trehalose in promoting resistance to osmotic stress and desiccation. We have opted to use the term “stress protectant” in an attempt to capture its function in maintaining protein and membrane structure during such stress. Although both roles as a stress protectant and an energy source have been described for trehalose, they have been evoked in different organisms and different contexts. Consequently, we do not believe it is trivial that it can simultaneously function in both capacities during starvation. Because we did not deem it trivial, we went to great lengths to show by multiple independent approaches that trehalose does support starvation survival as both a stress protectant and an energy source. Given the fat provisions dauer larvae are endowed with, their critical role as a dispersal stage, and their extreme resistance to a variety of stressors, it is not a surprise they rely on trehalose for these remarkable properties. However, it was unclear if survival of acute starvation as L1 larvae would be supported by trehalose other than as an energy source. Our results suggest that survival of starvation requires not only adequate calories but also preservation of macromolecular structure. We believe a fair number of readers will find this insight surprising. Finally, there is some evidence that starved L1 larvae have increased stress resistance compared to fed larvae (reviewed in Baugh, 2013). For example, it is routine to freeze starved but not fed larvae for long-term storage. Trehalose may contribute to increased stress resistance during L1 starvation, as suggested by our demonstration that it contributes to heat resistance.

Reviewer #2:This manuscript by Hibshman et al. approaches the unsolved mystery of how the daf-16/FOXO transcription factor promotes starvation resistance. It builds on the solid foundation of work by the senior author, Ryan Baugh. The conclusions of the paper are that gluconeogenesis and trehalose synthesis are regulated in a daf-16 dependent manner during starvation and that trehalose serves a dual role in starvation survival, as a fuel and as a biochemical starvation defense. The findings are novel, and lend new insight to the biology of starvation defenses. Given the importance of trehalose in insect species, one major criticism of the novelty of the findings is that of course it is well known that gluconeogenesis is critical for starvation defense in mammals, and further that FoxO is well known to regulate gluconeogenesis and transcription of genes encoding gluconeogenic enzymes in liver together with PGC-1alpha (as the authors note in the Discussion). Without evidence for a conserved role of trehalose across phylogeny to higher eukaryotes, this dampens enthusiasm slightly.Overall, the work is very rigorous and thorough. I similarly appreciate the use of pharmacologic and complementary genetic approaches. The measurement of corporal trehalose concentrations in supplemented worms makes the point very strongly that trehalose is the responsible moiety.1) Why does the number of genes differ between Figure 1 daf-16 starved vs. WT starved (650) and Figure 1+245 = 502)? Shouldn't this be the same 650 genes that distinguish both states?

The list of genes that were significant in Figure 1 differs from 1B due to background correction. That is, we only assessed genes that were detected in our experiment and included in the modENCODE data in 1B. Thus, we have a smaller list of significant genes to include in the overlap analysis.

2) Do the authors think that by doing the two-factor analysis to identify their 103 high stringency genes that they might have omitted important genes in the starvation response? For example, if a gene is "on" in both starvation and the fed state, and does not change in WT animals but daf-16 is necessary for its expression which is important in starvation, would that gene have been unfairly excluded? Asking for significant interactions with condition and genotype may have biased the expression analysis.

The reviewer is correct that the two-factor analysis is undoubtedly a conservative approach. Being conservative, it may have limited sensitivity with a gain in specificity, which is not expected to introduce bias. The example given would be detected as an interaction, assuming the effects described were sufficient to exceed associated measurement error. Qualitative examination of the data (e.g., in heat maps after hierarchical clustering; Figure 1—figure supplement 1) revealed the effect of *daf-16* to be largely conditional. That is, a majority of affected genes were up-regulated by starvation in WT but not *daf-16* mutants. Given our effort to capture immediate-early effects rather than broad, potentially indirect effects, we reasoned that the two-factor analysis would yield the most specific set of targets. However, we recognize that this method is likely to exclude some genes that function in the starvation response. Given the primary role of *daf-16* during starvation (expected based on prior knowledge and also supported by the results), and our particular interest in the starvation response, we also employed a standard pairwise analysis to identify genes that differ between *daf-16* and WT during starvation to increase sensitivity. We used the two-factor analysis in Figure 1 to arrive at carbon metabolism, then we used the pairwise comparison for our annotation of differentially expressed genes in Figure 1.

3) What time point is used to garner the 3.6 fold increase in trehalose in Figure 2? It seems much more blunted in Figure 2, is this because the change is relative to the same timepoint fed animals or relative to time = 0?

The trehalose measurements in Figure 2 were taken 24 hr after bleaching. This is equivalent to the 12 hr time point in Figure 2. The fold-change is calculated as the ratio of starved to fed at this same time point within samples. Unfortunately, there is some noise in the assay which likely accounts for variation across panels. There is a 6.4 fold increase in trehalose levels of starved worms compared to fed worms in the time course in Figure 2. We also realized that the fold changes reported in Figure 2 had not been updated to reflect the complete data set and these have been changed (from 3.6 to 3.9 for WT and 1.4 to 1.5 for *daf-16*).

4) While I appreciate the corroboration of genetic with pharmacologic data (Figure 3), is it known that 3-mercaptopicolinic acid inhibits gluconeogenesis in the worm as it does in mammalian hepatocytes?

We could not find a specific reference to 3-mercaptopicolinic acid being used in the worm. However, it is also functional in plants in addition to mammalian hepatocytes. We suspect that given the strong conservation of its target for inhibition, PEPCK, that it should function similarly in the worm. We also note that the 3-MPA result corroborates the mutant data in Figure 3, supporting the idea that 3-MPA works in worms as in other systems. We recognize that in publishing the use of 3-MPA as a gluconeogenic inhibitor in the worm we are setting a precedent and have therefore added extra clarification that we do not actually know how 3-MPA works in *C. elegans*.

5) It is very curious that trehalose supplementation reduces starvation-induced decrease in body length. Does trehalose supplementation also alter the rate of loss of protein or loss of triglyceride mass with starvation?

It is likely that there are other consequences of trehalose supplementation as suggested. It is reasonable to predict that other energy sources like triglycerides, glycogen, or even protein might be maintained if trehalose is present as an alternative fuel. While these are intriguing questions, we believe they are beyond the scope of this work.

6) The inability to rescue daf-16 starvation sensitivity (Figure 4) with trehalose, and the greater effect of trehalose in WT animals suggests that a "downstream blockade" prevents daf-16 mutants from utilizing trehalose effectively. The authors should address this, experimentally if possible. Perhaps forced expression of trehalase will permit more effective use of trehalose?

Yes, this is an astute observation! We interpret the result in Figure 4 to suggest that there are other effectors of *daf-16* beyond just trehalose synthesis. Likewise, disruption of trehalose synthesis does not completely abrogate increased starvation resistance of *daf-2* mutants. In addition, we show later in the manuscript that *tps-1* and *tps-2* are required to maintain high levels of corporeal trehalose even with supplementation, and these genes are not up-regulated during starvation in the *daf-16* mutant, potentially also limiting the effect of trehalose supplementation on *daf-16* mutants. These interpretations of the result are each included in the Discussion.

7) Why is the supplementation effect of 50mm trehalose so much more dramatic in Figure 5 versus Figure 4? Is the effect that noisy? It's almost 2x the effect in Figure 5 versus Figure 4.

Unfortunately, despite our best efforts there is variability in measuring starvation survival. We are not sure what the cryptic variables are, but we present all of our results in transparent fashion. We are careful to include new controls in every experiment and these are reported independently in different figure panels. Indeed, you were able to see the variation. We find it reassuring to see the same effects over and over again, at least qualitatively, and we test hypotheses by independent complementary approaches whenever possible to ensure reliability of our interpretations.

8) If trehalose supports starvation survival partially without playing a role as a metabolic fuel, supplementation of the quintuple tre mutant should have the same benefit from a,a trehalose versus a,b trehalose. Similarly, since it is not metabolized, the a,b trehalose supplementation should have the same effect in prolonging starvation survival whether 20mM 2-DG is present or not.

Agreed! These are predictions of our model. However, we believe these particular experiments to be beyond the scope of this work, as we have already supported the model by multiple independent lines of evidence and these experiments require substantial time and resources.

9) The very low levels of trehalose in the trehalose supplemented tps-1;tps-2 double mutant makes the strong argument that there is some cycling of 6-carbon sugars necessary for starvation survival. I'm not suggesting this experiment, but does inhibition of the tre genes in a tps-1;tps-2 double mutant restore trehalose levels upon supplementation?

We found the inference of cycling between sugars to be interesting though unexpected. And yes, this is another very intriguing experiment, but again we believe it is beyond the scope of this work.

10) More of a comment than a criticism – there is a phenomenon known as the glucose paradox, which shows that liver does not directly metabolize glucose to glycogen. Rather, lactate and pyruvate serve as fuels for glycogen synthesis, and any glucose that serves as a substrate for glycogen synthesis first has to go through three-carbon metabolism. This suggests two pools of glucose in the liver, that which is destined for glycolysis and that which is destined for glycogen synthesis. Is the equivalent true here? For example, are there two pools of glucose, that which is destined for glycolysis and that which is destined for trehalose synthesis? While not entirely analogous, a similar phenomenon could begin to explain the observation that it is not possible to maintain trehalose levels without tps-1;tps-2. The fact that glycolysis inhibition doesn't markedly affect starvation survival argues strictly against the likelihood that glucose must pass through three carbon metabolism to be subsequently made into trehalose.

Thank you for this very interesting comment. It is intriguing to consider that in our system there could be a similar phenomenon to the glucose paradox. That is, there could be two distinct pools of glucose – one for glycolytic catabolism and another for trehalose synthesis. We are also curious about the import mechanisms of trehalose and why it seemingly must be broken down to glucose and resynthesized. Perhaps this is a necessity based on intestinal uptake. While these are interesting ideas we decided not to incorporate them into the Discussion given the level of speculation involved.

Reviewer #3:Hibsham et al. present a very interesting study of the role of trehalose synthesis during starvation for C. elegans survival. This is a well-designed and timely study as trehalose has recently been proposed to enhance cryopreservation in C. elegans. The authors use a combination of transcriptomics, metabolomics, and physiological characterization in their study and although there is a significant amount of data presented, several aspects are underdeveloped and do not fully support the claims made.1) The authors study gluconeogenesis and trehalose synthesis based on their transcriptional profiling studies. The transcripts studies, although significantly different between growth conditions, are not impressive based on magnitude. This is perhaps because most of these enzymes are regulated at the protein level and not transcriptionally. In light of the timeframe of the response it is more likely that PTMs will play an important role. This should be examined, particularly since the glyoxylate shunt, gluconeogenesis and trehalose synthesis are previously documented in starvation and dauer, thus diminishing the novelty of the data presented.

The reviewer is absolutely correct that we completely failed to mention the likely significance of post-translational modifications in metabolic adaptation to starvation. As noted above, we have added a paragraph to the Discussion dedicated to post-translational modifications (subsection “*daf-16*/FoxO shifts central carbon metabolism during starvation”, second paragraph).

It is true that expression analysis has implicated the glyoxylate shunt, gluconeogenesis and trehalose synthesis in dauer larvae, but physiological significance of these changes outside of desiccation tolerance, and in particular for starvation resistance, had not been demonstrated.

2) The authors state that their findings represent a metabolic "Flux shift", but what is it a flux from? What pathway is being repressed in this case?

We suggest that similar to models of dauer metabolism, a “microaerobic” metabolic state prioritizes fat mobilization instead of storage, activity of the glyoxylate shunt instead of the TCA cycle and the electron transport chain, as well as a bias towards utilization of gluconeogenesis over glycolysis. We have more explicitly stated this in the discussion of our model.

3) What is the consequence of increased survival after L1 starvation with/without trehalose? Similarly, does trehalose protect from starvation at other developmental stages?

These are interesting questions. We expect that given the increased starvation survival of worms in the presence of trehalose, recovery from starvation will also be improved. There are relatively few exceptions that break the correlation between starvation survival and developmental fidelity upon recovery. Generally worms that remain healthier during starvation recover better. Thus, we do not believe this experiment would substantially strengthen our conclusions.

Though we have not specifically tested the effect of trehalose at other developmental stages, we suspect that we would see similar results. We believe a comparative study to characterize starvation responses across developmental stages, though potentially interesting, is beyond the scope of this manuscript. Our working model is that the acute response to starvation we are studying at the L1 stage is representative of acute starvation responses at other developmental stages. Our developmental analysis of starvation responses in later larval stages supports this view (Schindler et al., 2014, PloS Gen). We also note in the Discussion that others have reported transcriptional up-regulation of *icl-1*/isocitrate lyase/malate synthase, whose product mediates the glyoxylate shunt, during starvation in other stages, suggesting a common starvation response.

4) Lastly, the authors present a model for trehalose transport between organs (in the Abstract and Discussion), but there is no evidence of this presented in the data. This is an interesting aspect of this story in light of the organism level phenotypes, which should be supported by the data.

We agree that trehalose transport is an interesting idea in light of organismal phenotypes. Given lack of direct supporting evidence of transport, we have removed any suggestions that we provide data to support this conclusion, while also removing mention of it from the Abstract and other summaries. We have kept discussion of this possibility as speculation in the Discussion, potentially spurring future research of this compelling hypothesis.